# A modeling study of temporal and spatial $p$CO$_2$ variability on the biologically active and temperature-dominated Scotian Shelf

Krysten Rutherford[1], Katja Fennel[1], Dariia Atamanchuk[1], Douglas Wallace[1], and Helmuth Thomas[1,2]

[1]Department of Oceanography, Dalhousie University, 1355 Oxford Street, Halifax, Nova Scotia, B3H 4R2, Canada
[2]Institute of Coastal Research, Helmholtz Center Geesthacht, D-215502 Geesthacht, Germany

**Correspondence:** Krysten Rutherford (krysten.rutherford@dal.ca)

**Abstract.** Continental shelves are thought to be affected disproportionately by climate change and are a large contributor to global air-sea carbon dioxide (CO$_2$) fluxes. It is often reported that low-latitude shelves tend to act as net sources of CO$_2$ whereas mid- and high-latitude shelves act as net sinks. Here, we combine a high-resolution regional model with surface water time series and repeat transect observations from the Scotian Shelf, a mid-latitude region in the northwest North Atlantic, to
determine what processes are driving the temporal and spatial variability of partial pressure of CO$_2$ ($p$CO$_2$) on a seasonal scale. In contrast to the global trend, the Scotian Shelf acts as a net source. Surface $p$CO$_2$ undergoes a strong seasonal cycle with an amplitude of $\sim$200-250 $\mu$atm. These changes are associated with both a strong biological drawdown of Dissolved Inorganic Carbon (DIC) in spring (corresponding to a decrease in $p$CO$_2$ of 100-200 $\mu$atm), and pronounced effects of temperature, which ranges from 0$^o$C in the winter to near 20$^o$C in the summer, resulting in an increase in $p$CO$_2$ of $\sim$200-250 $\mu$atm. Throughout the
summer, events with low surface water $p$CO$_2$ occur associated with coastal upwelling. This effect of upwelling on $p$CO$_2$ is also in contrast to the general assumption that upwelling increases surface $p$CO$_2$ by delivering DIC-enriched water to the surface. Aside from these localized events, $p$CO$_2$ is relatively uniform across the shelf. Our model agrees with regional observations, reproduces seasonal patterns of $p$CO$_2$, and simulates annual outgassing of CO$_2$ from the ocean of $+1.7 \pm 0.2$ mol C m$^{-2}$ yr$^{-1}$ for the Scotian Shelf, net uptake of CO$_2$ by the ocean of $-0.5 \pm 0.2$ mol C m$^{-2}$ yr$^{-1}$ for the Gulf of Maine, and uptake by
the ocean of $-1.3 \pm 0.3$ mol C m$^{-2}$ yr$^{-1}$ for the Grand Banks.

## 1 Introduction

The global ocean acts as a major sink of CO$_2$ from the atmosphere (e.g., Le Quéré et al., 2018; Gruber et al., 2019; Landschützer et al., 2014; Rödenbeck et al., 2015), but it has been suggested that flux density (or flux per unit area) on continental shelves is larger than in the open ocean (Chen et al., 2013; Laruelle et al., 2014). Therefore, compared to their size, continental shelves are
thought to disproportionately contribute to global air-sea CO$_2$ fluxes (Laruelle et al., 2010). Additionally, they are susceptible to climate change on much shorter timescales than the open ocean (Cai et al., 2010) and are experiencing increasing impacts of human activity (Cai, 2011; Doney, 2010; Gruber, 2015). Given their high susceptibility to negative impacts from climate change, and their potentially significant contribution to global air-sea CO$_2$ fluxes, it is important to understand the drivers underlying inorganic carbon dynamics on continental shelves.

It is generally thought that continental shelves at mid- to high-latitudes act as net sinks of atmospheric $CO_2$ while those at low latitudes act as net sources (e.g. Chen and Borges, 2009; Cai et al., 2006; Laruelle et al., 2014; Roobaert et al., 2019). There are, however, notable deviations from this global-scale pattern. The Scotian Shelf, a mid-latitude shelf off the coast of eastern Canada, is one example with large discrepancies between independent estimates of air-sea $CO_2$ flux (Fennel et al., 2019). Direct measurements made using a moored CARIOCA buoy on the Scotian Shelf indicate that the shelf acts as a net source of $CO_2$ to the atmosphere (Shadwick et al., 2010, 2011; Shadwick and Thomas, 2014). These findings are in contrast to other studies using observations from the SOCAT database, indicating that the Scotian Shelf follows the global trend and acts as a net sink of $CO_2$ (Laruelle et al., 2014, 2015; Signorini et al., 2013). These contrasting results for the Scotian Shelf emphasize the large uncertainty inherent in shelf-wide $CO_2$ flux estimates.

Continental shelves are highly complex and dynamic regions where many biological and physical processes modulate $CO_2$ flux (Laruelle et al., 2014, 2017; Roobaert et al., 2019). The partial pressure of $CO_2$ ($pCO_2$) in the ocean is one of the key factors which determines the air-sea $CO_2$ flux. Recent global studies found that thermal controls dominate the seasonality of $pCO_2$ but that these alone cannot describe observed $pCO_2$ variations, particularly in temperate and high latitudes (Roobaert et al., 2019). High rates of primary production on continental shelves (Chen and Borges, 2009) are another important driver of seasonal changes in $pCO_2$.

Continental margins are also subject to intense horizontal transport processes, which act as additional drivers of $CO_2$ fluxes. For example, the Continental Shelf Pump, a term first coined by Tsunogai et al. (1999) in relation to the East China Sea, describes the movement of shelf water high in dissolved inorganic carbon (DIC) across the shelf break to the subsurface open ocean leading to an influx of atmospheric $CO_2$. This mechanism is thought to mainly occur at mid- to high-latitude shelves since it relies on winter cooling to create dense shelf water that is transported to the open ocean's subsurface layers. Upwelling is another well-studied transport mechanism driving shelf-wide $CO_2$ dynamics. The California Current system is a typical example of an upwelling system (Chavez et al., 2017; Hickey, 1998; Fennel et al., 2019; Feely et al., 2008). Here, winds drive coastal upwelling, which brings DIC-rich water to the surface along the continental shelf and creates favourable conditions for $CO_2$ outgassing to the atmosphere.

Altogether, these complex shelf dynamics lead to large spatial and temporal variability of $pCO_2$ (Previdi et al., 2009). Such large variability combined with limited data availability for many continental shelves make it difficult to accurately constrain $CO_2$ fluxes. Limited data availability in space and time, often with seasonal biases, is a prime source of uncertainty in flux estimates that can only be overcome with more uniformly distributed sampling. To fully capture how ocean margins are reacting to perturbations caused by the steady input of anthropogenic $CO_2$ to the atmosphere, it is important to understand the processes underlying both spatial and temporal evolution of shelf-wide $pCO_2$.

Numerical models can be useful when investigating such complex interactions and constraining $CO_2$ flux since they can interpret sparse measurements through the mechanistic representations of relevant processes. In the present study, we employ a high-resolution biogeochemical model of the northwest North Atlantic to examine the magnitude, variability, and sign of the air-sea $CO_2$ flux on the Scotian Shelf. Previous studies have evaluated our model's ability to represent the physical (Brennan et al., 2016; Rutherford and Fennel, 2018) and biological (Laurent et al., 2021) dynamics of the region. Here, we focus solely

on the model representation of inorganic carbon dynamics, especially the spatial and temporal variability of $p$CO$_2$ on a seasonal scale on the Scotian Shelf in light of new, high-resolution, shelf-wide observations.

Our overall goal is to show how both biological and transport processes work together seasonally on the Scotian Shelf to set shelf-wide surface $p$CO$_2$. We additionally discuss event-based variability of the air-sea CO$_2$ flux, and, especially, how short-term, upwelling-favourable wind events throughout the summer create spatial variability of CO$_2$ on the Scotian Shelf. To

65 accomplish these goals, our paper: (1) discusses the seasonal cycle of $p$CO$_2$ across the shelf; (2) investigates the spatial variability of $p$CO$_2$, particularly during the summer months, and (3) reports shelf-wide air-sea CO$_2$ flux estimates in comparison to previously reported estimates. We discuss the importance of our findings in terms of global patterns of air-sea CO$_2$ flux and carbon cycling.

## 2   Study Region

The Scotian Shelf (Figure 1) is uniquely located at the junction of the subpolar and subtropical gyres (Loder et al., 1997; Hannah et al., 2001). Regional circulation is dominated by southward transport of the Labrador Current (Loder et al., 1998; Fratantoni and Pickart, 2007). As a result, cool Arctic-derived water accumulates along the northwestern North Atlantic continental shelf separating fresh shelf waters from warmer and salty slope waters (Beardsley and Boicourt, 1981; Loder et al., 1998; Fratantoni and Pickart, 2007).

The Scotian Shelf in particular is controlled by inshore and shelf-break branches of the southwestward moving current. The shelf-break branch inhibits the movement of water across the shelf break of the Scotian Shelf (Rutherford and Fennel, 2018). As a result, water moves predominantly along-shelf so that residence times in the region are relatively long, with water being retained on the Scotian Shelf for an average of 3 months before moving further southwest on the shelf (Rutherford and Fennel, 2018). In terms of vertical structure, the Scotian Shelf shifts between a two-layer system in the winter, when a cold, fresh layer

sits over a warm, salty deep layer, and a three-layer system in the spring and summer, when a warm surface layer forms in the top 20 m above the cold intermediate layer between 20–100 m, and the warm and salty deep layer (Dever et al., 2016).

The Scotian Shelf is additionally characterized by a large, shelf-wide spring bloom initiated in late-March (Ross et al., 2017; Fournier et al., 1977; Mills and Fournier, 1979), when the mixed layer is still relatively deep and temperature is at its coldest (Craig et al., 2015). The initiation of the spring bloom in late March has rapid and large impacts on the observed $p$CO$_2$

seasonality (Shadwick et al., 2010, 2011).

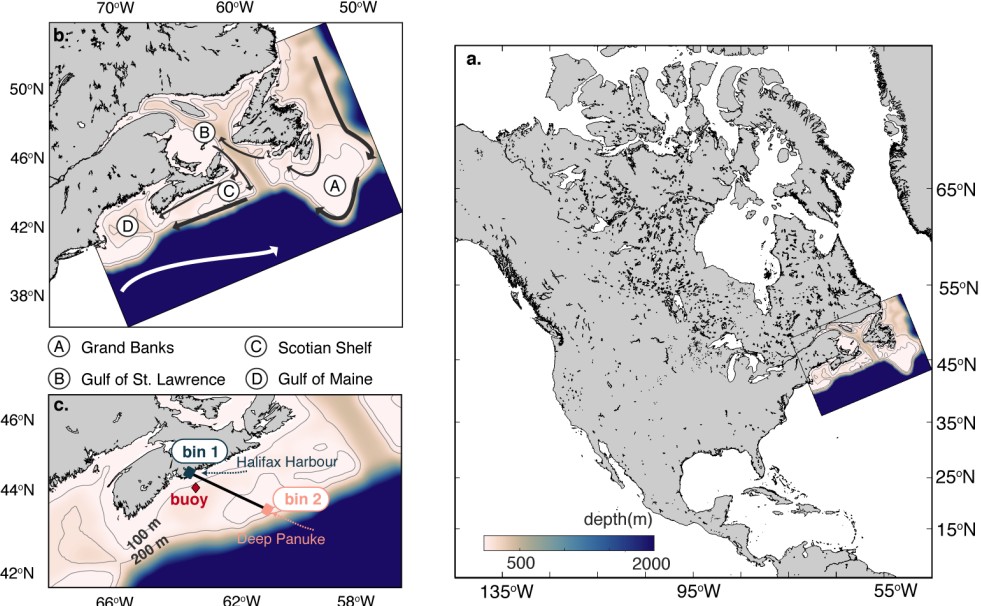

**Figure 1.** Bathymetric maps of the model domain. (a) Map of North America, including the location of the model domain. (b) A zoomed-in map of the model domain with mean current locations. (c) Zoomed-in map of the Scotian Shelf, and indicates the location of the CARIOCA buoy (red diamond) and the Atlantic Condor Transect (black line). Bin 1 (Halifax Harbour) and bin 2 (Deep Panuke) are used for analyses of spatial variability. All maps show the 100 m and 200 m isobaths.

## 3 Methods

### 3.1 Model setup and initialization

#### 3.1.1 Physical model setup

We employ a biogeochemical model, based on Fennel et al. (2006), Fennel and Wilkin (2009), and Laurent et al. (2021) that
is part of the Regional Ocean Modelling System (ROMS, v.3.5; Haidvogel et al., 2008). The physical model implementation, described in more detail in Brennan et al. (2016), has 30 vertical levels and approximately 10 km horizontal resolution (240x120 horizontal grid cells), uses the GLS vertical mixing scheme (Umlauf and Burchard, 2003; Warner et al., 2005), atmospheric surface forcing from the European Centre for Medium-Range Weather Forecasts (ECMWF) global atmospheric reanalysis (Dee et al., 2011), and the "high-order spatial interpolation at the middle temporal level" (HSIMT) advection scheme for
tracers (Wu and Zhu, 2010). Physical initial and boundary conditions are defined using the regional physical ocean model of the northwest North Atlantic of Urrego-Blanco and Sheng (2012). Temperature and salinity are nudged towards the climatology of Geshelin et al. (1999) in a 10-grid-cell-wide buffer zone along open boundaries. Nudging strength decays linearly away from the boundaries to a value of zero in the 11th grid cell from the boundary. Tides are imposed from Egbert and Erofeeva. Climatological river discharge is imposed for 12 major rivers and uses observed long-term monthly means from Water Survey

Canada. Full details on the physical model setup and its validation can be found in Brennan et al. (2016) and Rutherford and Fennel (2018). These studies have shown that our model simulates the vertical structure and seasonal cycling of temperature and salinity on the shelf well. The model captures mesoscale features and the coastal upwelling events, and simulates the volume transport throughout the region in agreement with observation-based estimates.

### 3.1.2 Biogeochemical module

The biogeochemical model is based on the nitrogen-cycle model with inorganic carbon component of Fennel et al. (2006) and Fennel and Wilkin (2009) but was recently expanded to include 2 phytoplankton and 2 zooplankton functional groups (Laurent et al., 2021). For a detailed description and validation of the biological model, we refer to Laurent et al. (2021), who compared the model output with glider transects of temperature, salinity, and chlorophyll and in situ measurements of chlorophyll and nitrate. The model was evaluated on a seasonal scale for the entire model domain, mainly in the surface (top 100 m). Laurent
et al. (2021) showed that the model outperforms global models for the region for all variables and that the timing of the spring bloom is well represented, but the model slightly underestimates the magnitude of the bloom and tends to overestimate nitrate throughout the year.

For calculating air-sea $CO_2$ flux, according to the carbonate chemistry model of Zeebe and Wolf-Gladrow (2001), we use dissociation constants (K1 and K2) from Millero (1995) using Mehrbach et al. (1973) data on the seawater scale which are
115 deemed appropriate for the typical salinity ranges from 27 to 36.6 in the model domain (lower salinities are highly localized in the Gulf of St. Lawrence Estuary). Atmospheric $p$CO$_2$ is set to the seasonal cycle and secular trend derived from Sable Island monitoring data contributed by Environment Canada's Greenhouse Gas Measurement Program (Environment and Climate Change Canada, 2017). The long term linear trend in the atmospheric $p$CO$_2$ is $\sim$+2 $\mu$atm year$^{-1}$ (see Supplement for the full trend equation and figure). $CO_2$ solubility is calculated with the Weiss (1974) formulation. The gas transfer coefficient of
120 Ho et al. (2006) is used and depends on wind speed at 10 m above the sea surface and the Schmidt number. Further details of the biogeochemical model, including the carbonate chemistry equations, can be found in Laurent et al. (2017, Supporting Information). Carbon initialization, boundary conditions, and climatological nudging are calculated from relationships with temperature and salinity determined from bottle data for the region. DIC is nudged in an 80-grid-cell-wide buffer zone along the eastern boundary, with nudging linearly decaying away from a nudging timescale of 60 days at the boundary to a value of
125 0 in the 81st grid cell. At all other boundaries, a 10-grid buffer zone is used, as with temperature and salinity. Use of a wider boundary nudging zone along the eastern boundary was found to be beneficial in imposing low-frequency variability from the Labrador Sea at the northeastern boundary. The nudging zones are not used in the analysis.

Nitrate concentrations in rivers are prescribed from Global NEWS model output Seitzinger et al. (2005). DIC and total alkalinity (TA) in rivers were calculated by fitting a linear relationship with salinity from Gulf of St. Lawrence bottle data and
130 extrapolating to river water salinity. The model is initialized on January 1, 1999, from Urrego-Blanco and Sheng's (2012) solution for temperature and salinity. Nitrate ($NO_3^-$) concentrations are initialized from the regional climatologies as in Laurent et al. (2021). DIC and TA initial and boundary conditions were created from observationally based relationships with temperature (T) and salinity (S) using bottle data from regional cruises from 1997-2011 encompassing as far south as the Gulf of Maine and

as far north as the Labrador Sea (observations from DFO's AZMP program, see: dfo-mpo.gc.ca/science/data-donnees/azmp-pmza/index-eng.html#publications). Initialization relationships used only observations from December, January, and February (TA = 43S + 800, $r^2$ = 0.96; DIC = 1153 – 21.6T + 29.1S – 0.41$T^2$ + 0.63ST, $r^2$ = 0.90). Boundary conditions used observations that encompassed the entire year (TA = 41S + 875, $r^2$ = 0.92; DIC = 912.6 – 2.5T + 35.7S – 0.45$T^2$ + 0.12ST, $r^2$ = 0.80). The model is run for 16 years (1999-2014) with daily output. The present study analyses the model output from 2006-2014, with focus on year 2006. See the Supplement for a comparison of surface $p$CO$_2$ throughout the simulation and a brief validation of TA and DIC.

### 3.1.3 Taylor decomposition of upwelling events

To better understand the effects of coastal upwelling on surface $p$CO$_2$, we perform a Taylor Decomposition on the model output during one of the upwelling events focused on in this study, following a similar methodology to Rheuban et al. (2019), and Hauri et al. (2020). Here, we investigate the influence of T, S, DIC, and TA on $p$CO$_2$ following the equation

$$pCO_2 = f(T, S, DIC, TA) \tag{1}$$

where $f$ indicates the CO2SYS set of equations. We calculated anomalies, $\Delta$ $p$CO$_2$, from a reference value, $pCO_{2,0}$:

$$\Delta pCO_2 = pCO_2 - pCO_{2,0} \tag{2}$$

The reference values for each variable were calculated as the average of that variable along the Condor transect (see Figure 1) in the upper 40 m (i.e., the part of the water column affected by the upwelling event). We decomposed $\Delta$ $p$CO$_2$ relatively simply into perturbations related to T, S, DIC, and TA calculated as follows:

$$\Delta pCO_{2,T} = f(T, S_0, DIC_0, TA_0) - pCO_{2,0} \tag{3}$$

$$\Delta pCO_{2,S} = f(T_0, S, DIC_0, TA_0) - pCO_{2,0} \tag{4}$$

$$\Delta pCO_{2,DIC} = f(T_0, S_0, DIC, TA_0) - pCO_{2,0} \tag{5}$$

$$\Delta pCO_{2,TA} = f(T_0, S_0, DIC_0, TA) - pCO_{2,0} \tag{6}$$

We refer the reader to Rheuban et al. (2019) for a more detailed description of the Taylor Decomposition methodology.

### 3.2 Observational datasets

The moored CARIOCA buoy was located at Station 2 on the Halifax Line. Station 2 (HL2; 44.3$^o$N, 63.3$^o$W) is located about 30 km offshore from Halifax, Nova Scotia, and occupied monthly by Bedford Institute of Oceanography. The buoy measured surface water (at approximately 1 m depth) temperature, conductivity, $p$CO$_2$, salinity, and Chl-a fluorescence every hour and was deployed from 2007 to 2014 with several gaps in data due to calibration and maintenance (see Table S1 in Supplement). $p$CO$_2$ was estimated using an automated spectrophotometric technique (Lemay et al., 2018). The raw $p$CO$_2$ data contained high-amplitude spikes, with increases from 400 $\mu$atm to over 1000 $\mu$atm within a few hours, which were measuring artifacts

and did not represent $pCO_2$ of surrounding water. These spikes were removed by binning all years of the $pCO_2$ observations into a 365-day of year (DOY) seasonal cycle. Any points that were outside 1.5 standard deviations of the 1-month moving average $pCO_2$ were discarded. This method removed only the extreme values and maintained much of the observed variability (see Figure 2).

The sensor-based underway system, Dal-SOOP (Arruda et al., 2020), was installed on the multipurpose platform supply vessel Atlantic Condor (operated by Atlantic Towing Ltd.) and has been measuring a suite of biogeochemical parameters, including $pCO_2$, in the surface water since May 2017. The ship transits weekly to biweekly between the Halifax Harbour (Bin 1) and the Deep Panuke gas platform off Sable Island (Bin 2) on the Scotian Shelf (Figure 1). The Atlantic Condor $pCO_2$ data underwent standard QA/QC procedures, which included pre-, post-deployment and regular zero-calibration of the $pCO_2$ sensor (Pro-Oceanus Inc, Canada) and associated data corrections. The QC'd data has been deposited into the Surface Ocean $CO_2$ Atlas (SOCAT v.2020), where it was attributed an accuracy of $\pm$ 10 $\mu$atm. Performance of the novel Dal-SOOP system was assessed during a 2-month transatlantic cruise in comparison with a conventional $pCO_2$ equilibrator and showed good agreement with the latter (i.e. -5.7 $\pm$ 4.0 $\mu$atm;  Arruda et al., 2020).

During the QC/QA procedure, some data collected in close proximity to Halifax, and corresponding to the outbound transects, were removed. Some of these data were biased high and attributed to prolonged ship layover in port allowing for a build-up of high $pCO_2$ within the Dal-SOOP system due to respiration. The active pumping that delivers fresh seawater to the measurement system is triggered by a GPS signal when the ship leaves the harbour; as a result, there can be a delayed response from the $pCO_2$ sensor to the much lower $pCO_2$ signals observed immediately outside the harbour. To account for the bias, values that were 2 standard deviations from the mean $pCO_2$ value for the latitudinal bin closest to the Halifax Harbour were removed for some transects. Only three transects were removed.

The CARIOCA and Atlantic Condor transect observations were mapped onto year 2006 for comparison directly with this year in the model using the linear trend in atmospheric $pCO_2$ (+ 2 $\mu$atm year$^{-1}$). Where numbers are reported comparing the model mean to observations, the observations were mapped to year 2010 (the median year of our model simulation). For comparison of the modelled flux to the flux estimates from the CARIOCA buoy, years 2006-2014 in the model were used and no mapping of the observations was performed.

## 4 Results

### 4.1 $CO_2$ time series and transect

Both the model and observations at the CARIOCA buoy location (see Figure 1) are shown as a seasonal cycle in Figure 2 (chlorophyll, $pCO_2$, temperature, and temperature-normalized $pCO_2$). The buoy observations show a distinct and recurring seasonal cycle in $pCO_2$. Specifically, $pCO_2$ slightly decreases (from $\sim$450 to 425 $\mu$atm) from day 0 to 75. In late March, at approximately day 75, there is a large (100-200 $\mu$atm) and rapid (over $\sim$25 days) drop of $pCO_2$ associated with DIC drawdown due to the spring bloom (the dashed line indicates the peak in chlorophyll and its alignment with the lowest $pCO_2$ value). This drawdown of DIC occurs while the surface temperature is relatively constant and at its annual minimum.

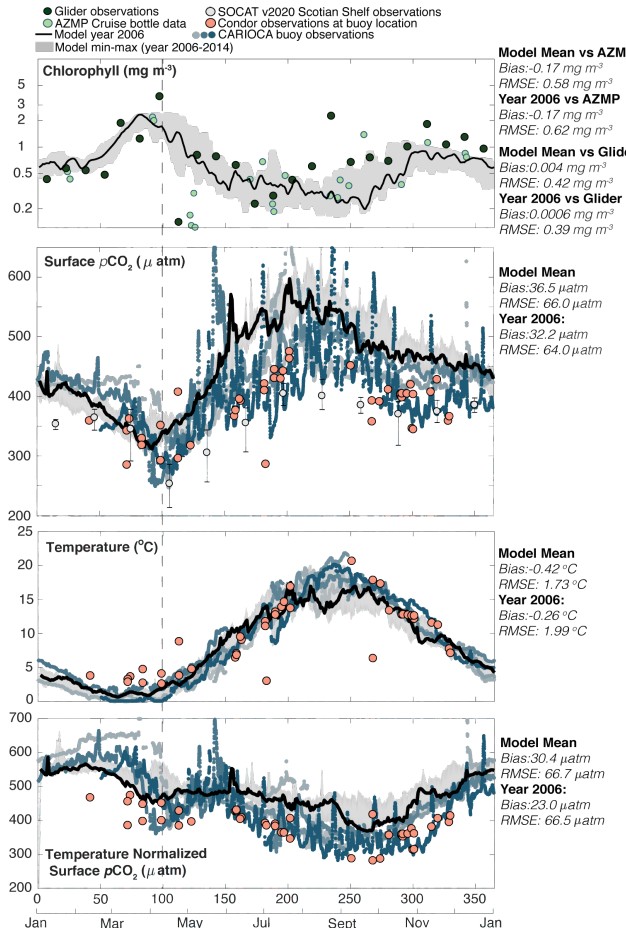

**Figure 2.** Seasonal (from top to bottom, with RMSE and bias in reference to year 2006) (a) chlorophyll (Glider: RMSE: 0.39 mg m$^{-3}$, bias: 0.0006 mg m$^{-3}$; AZMP: RMSE: 0.62 mg m$^{-3}$, bias: -0.17 mg m$^{-3}$); (b) $pCO_2$ (RMSE: 64.0 $\mu$atm , bias: 32.2 $\mu$atm ); (c) temperature (RMSE: 1.99 $^o$C, bias: -0.26 $^o$C); (d) temperature-normalized $pCO_2$ following Takahashi et al. (2002) (RMSE: 66.5 $\mu$atm, bias: 23.0 $\mu$atm) at STN 2 on the Scotian Shelf. The model year 2006 is shown with the thick black line and min-max in the model from years 2006-2014 with the grey shaded area in all panels. In (a) the dark green points are AZMP bottle data and light green points are glider data. In (b-d) observations from the moored CARIOCA buoy are shown as small blue points, with lighter shades of blue indicating earlier observations and darker shades indicating more recent observations, and observations from the Atlantic Condor transects at approximately the same location as the buoy are shown in large pink points. Both the Condor and CARIOCA buoy observations are mapped to year 2006 using the atmospheric trend in $pCO_2$. Light grey points are monthly mean SOCAT observations for the entire Scotian Shelf and the error bars are the 10th and 90th percentiles.

Following the drop in $pCO_2$ associated with the spring bloom, around day 100, surface water starts to warm, and this warming dominates the $pCO_2$ seasonal cycle with a maximum value of approximately 450–500 $\mu$atm reached around day 200-250 (mid to late summer). Around day 250, temperatures and $pCO_2$ start to decrease. Also shown is the temperature-normalized $pCO_2$

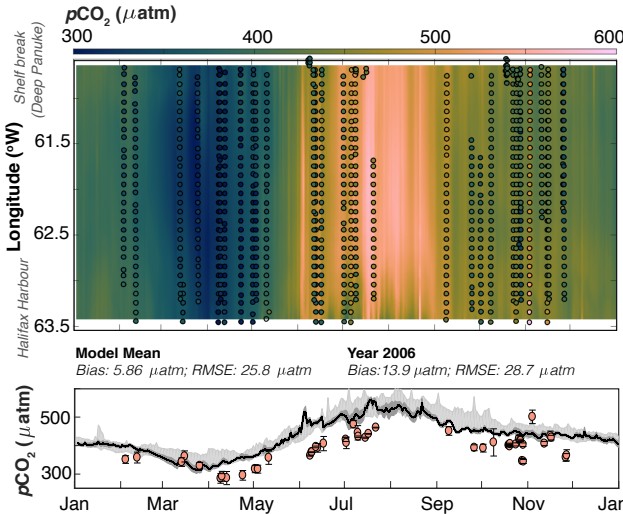

**Figure 3.** Model-data comparison along the Atlantic Condor transect. The top panel shows $pCO_2$ (in colour) evolving over time (x-axis) along the transect (longitude on the y-axis; Halifax Harbour to shelf break). The background is the model average $pCO_2$ along the transect and the points are the Atlantic Condor data binned into $0.1^o$ longitudinal bins. The bottom panel shows the average $pCO_2$ along the transect (y-axis) as it evolves over the seasonal cycle (x-axis). The line is year 2006 from the model averaged across the transect, the dark grey shaded area is the standard deviation, and the light grey shaded area is the min-max $pCO_2$ along the transect from 2006-2014. The points are the average and the error bars are standard deviation of observational $pCO_2$ across each transect. The Condor observations are mapped to year 2006 in both panels using the atmospheric trend in $pCO_2$. RMSE: 28.7 $\mu$atm; Bias: 13.9 $\mu$atm.

using the Takahashi et al. (2002) method for removing the thermal component of $pCO_2$ variations. The biological drawdown of DIC is visible in the temperature-normalized $pCO_2$ during the spring bloom starting around day 75 and a further decline throughout summer from day 150 to 250. This indicates that the overall increase in the non-normalized $pCO_2$ in summer is driven by increasing temperatures, and that biological processes tend to draw down DIC during this period.

Most of the Atlantic Condor observations at this location fall within the envelope of the buoy observations' $pCO_2$ seasonal cycle. The monthly mean SOCAT v2020 $pCO_2$ for the entire Scotian Shelf also falls within the spread of buoy observations for most months. Exceptions include February and August, when the SOCAT observations are lower than the buoy observations, and September and October, when the SOCAT observations are at the low end of the buoy observations.

In terms of quantitative metrics, the model (year 2006) at the buoy location has an overall bias of 32.2 $\mu$atm and RMSE of 64.0 $\mu$atm compared to the buoy data. The model underestimates $pCO_2$ throughout January and February (day 0-80) partly because its spring bloom starts earlier than in the observations. The bloom-related minimum in $pCO_2$ in the model is approximately 50-75 $\mu$atm higher than the buoy observations and approximately 25–50 $\mu$atm higher than the Atlantic Condor observations. Temperature then dominates the $pCO_2$ seasonality in the model over a similar period as in the observations. During the summer (day 150-300), the model overestimates $pCO_2$ but follows a similar cycle as the observations throughout the

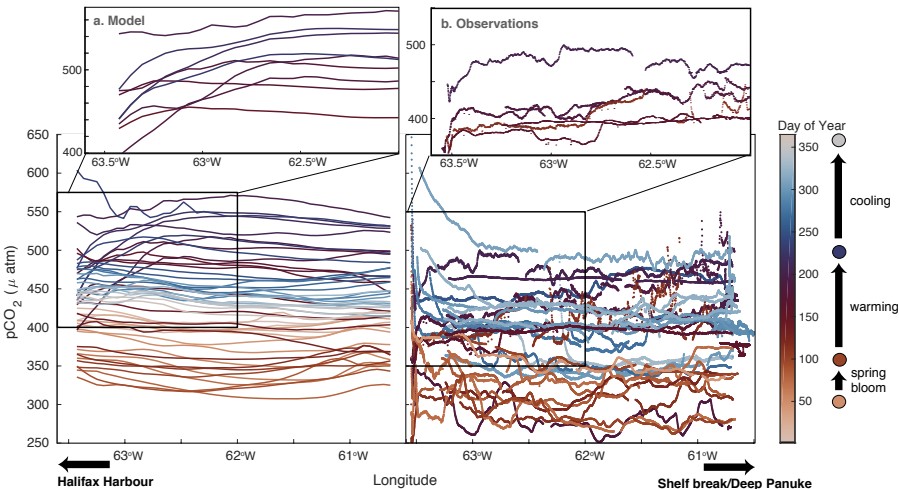

**Figure 4.** Temporal evolution of $p\text{CO}_2$ across the Atlantic Condor transect. X-axis is longitude, with the Halifax Harbour indicated on the left-hand side and the shelf break indicated on the right-hand side of each panel; Y-axis is $p\text{CO}_2$; and the colour indicates the day of the year. The left panel is year 2006 of the model along the transect every 7 days. The right panel are all of the observations along the transect. The upper insets zoom in on the indicated boxes showing only the events with lower $p\text{CO2}$ nearshore in the summer months (dark red/purple cololoured lines).

remainder of the year. The temperature-normalized $p\text{CO}_2$ has similar biases (underestimation from day 0-80; overestimation from day 150-300), an RMSE of 66.5 $\mu$atm, and an overall bias of 23.0 $\mu$atm for year 2006.

A comparison of simulated $p\text{CO}_2$ with the Atlantic Condor Transect observations along the average ship track (Figure 1) is shown in Figure 3. Compared to the Atlantic Condor observations, the model (year 2006) has a bias of 13.9 $\mu$atm and an RMSE of 28.7 $\mu$atm. The model tends towards slightly higher $p\text{CO}_2$ across the shelf compared to the ship data, but the bias along the ship track is about half the magnitude as that at the buoy. The seasonal cycle along the ship track (Figure 3) is similar to that at the buoy (Figure 2). The top panel of Figure 3 shows qualitatively good agreement between the model and observations across

the whole transect, which is reflected in the averaged $p\text{CO}_2$ in the bottom panel. The model does a very good job at representing $p\text{CO}_2$ throughout the winter (November through March) but does not reproduce the full spring bloom drop in $p\text{CO}_2$ across the whole shelf throughout April as observed. The model also overestimates $p\text{CO}_2$ throughout most of June and July. The seasonal cycle across the transect is relatively uniform throughout most of the year; however, there are some exceptions, for example, throughout July $p\text{CO}_2$ is relatively low near the shelf break in both the model and observations.

**4.2   Effects of upwelling events**

To better understand the effect of physical events on shelf-wide $p\text{CO}_2$, this section focuses on the cross-shelf variations in year 2006. Figure 4 shows the evolution of $p\text{CO}_2$ along the Atlantic Condor transect throughout the year in both model (Figure 4a) and observations (Figure 4b). As in Figure 2 and Figure 3, the seasonal cycle of $p\text{CO}_2$ extends across the entire shelf. Starting

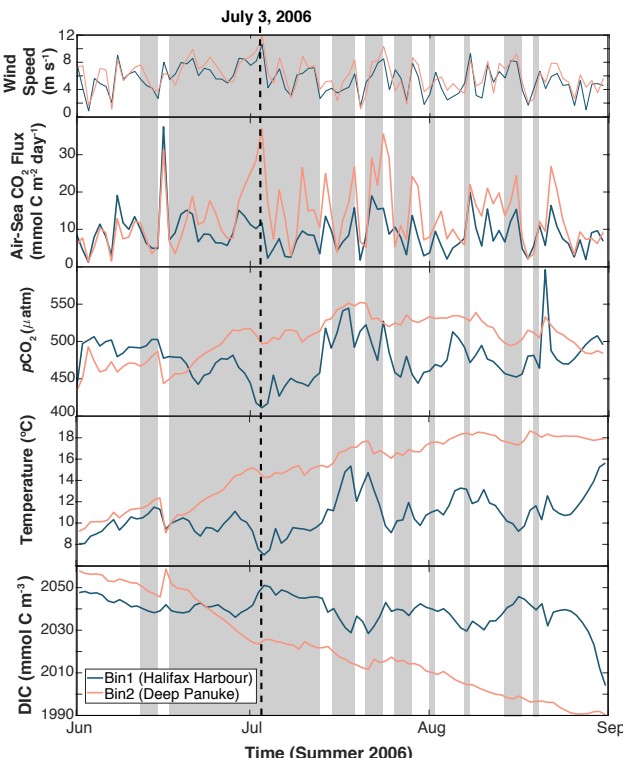

**Figure 5.** Time series of variables in two bins along the Condor Transect (see Figure 1) during summer 2006. From top to bottom: (a) wind speed, (b) air-sea $CO_2$ flux, (c) $pCO_2$, (d) temperature and (e) dissolved inorganic carbon (DIC). Shaded area indicates when there was upwelling-favourable winds nearshore (Bin 1). The blue lines indicate the values from the nearshore bin closest to the Halifax Harbour and the pink lines indicate values from the offshore bin near the Deep Panuke oil platform.

in January (light beige), $pCO_2$ is around 400 $\mu$atm. In March ($\sim$day 50; golden orange colour), $pCO_2$ starts to decrease,
reaching a minimum of approximately 325 $\mu$atm in the model and around 275-300 $\mu$atm in the observations (day 100; dark brown colour). $pCO_2$ subsequently increases again due to warming in the late spring/early summer and reaches a maximum of about 550 $\mu$atm in the model and 525 $\mu$atm in the observations (day 200; purple values). Following this peak in $pCO_2$, both the model and observations start to decline, associated with cooling (days 225 to 325; purple to light blue). Small-scale spatial variability in the observations is not captured by the model, but may, at least in part, be due to measurement artifacts of the
underway system.

The insets in Figure 4 highlight events in summer (purple) in the northwestern half of the transect closest to Halifax, when $pCO_2$ decreases by 50–100 $\mu$atm within $\sim$40 km off the coast in the model and approximately 25 km off the coast in the observations. With more obvious examples in the model than in the observations, we use the model to investigate into a possible explanation for this decreased $pCO_2$. Figure 5 highlights the differences in $pCO_2$, air-sea $CO_2$ flux, temperature, and
DIC between two longitudinal bins along the Atlantic Condor transect throughout summer 2006 in the model. The bin locations

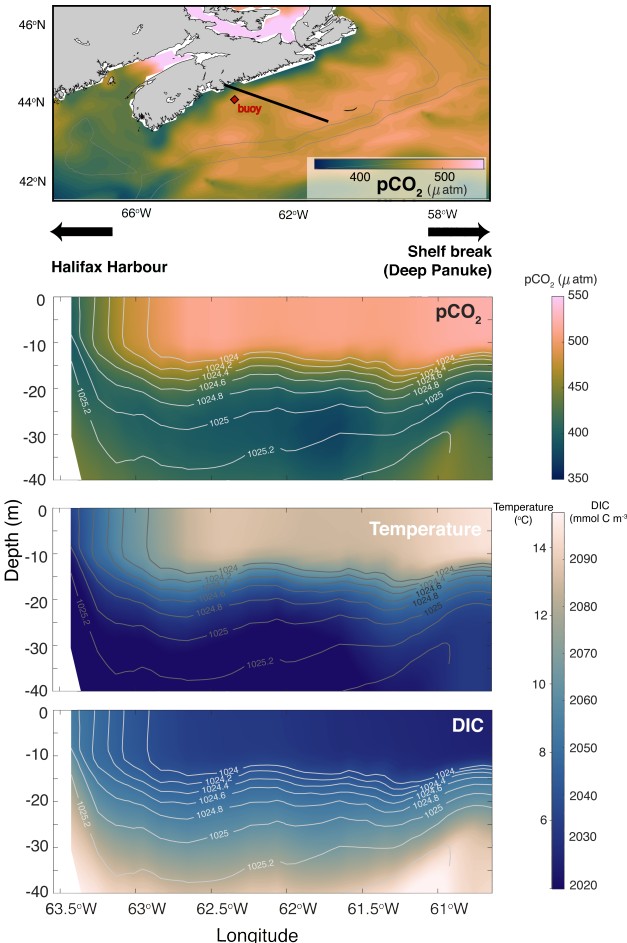

**Figure 6.** Surface map of $p\text{CO}_2$ (top panel), and transects along the average Atlantic Condor ship track of (top to bottom) $p\text{CO}_2$, temperature, and dissolved inorganic carbon (DIC) from the model taken during an upwelling event (Jul 3, 2006; see Figure 5). Contours in the transects are density. The top panel indicates the Condor transect with the black line and the location of the CARIOCA buoy with the red diamond.

are shown in Figure 1 and contrast data closest to the coastline (Halifax Harbour bin, $63.5^o$W to $63^o$W; blue) with data closest to the shelf break (Deep Panuke bin, $61^o$W to $60.5^o$W; pink). In the model throughout June to August 2006, there are low $p\text{CO}_2$ events nearshore corresponding to low temperature which occurs during upwelling-favourable winds. During some of these events, temperature nearshore is about $7^o$C lower than near the shelf break. These upwelling events and the subsequent

low $p\text{CO}_2$ signal result in a short-term lowering of air-sea $\text{CO}_2$ fluxes nearshore (blue) compared to farther offshore (pink) throughout the summer (at approximately half the flux value nearshore versus offshore throughout July).

The top panel in Figure 6 shows a snapshot of surface $p\text{CO}_2$ from the model during one of the upwelling events (July 3, 2006; vertical dashed line in Figure 5). $p\text{CO}_2$ is relatively uniform across most of the shelf. However, in a narrow band along the coastline, $p\text{CO}_2$ values are nearly 100 $\mu$atm lower than the rest of the shelf. The bottom panels in Figure 6 show transects

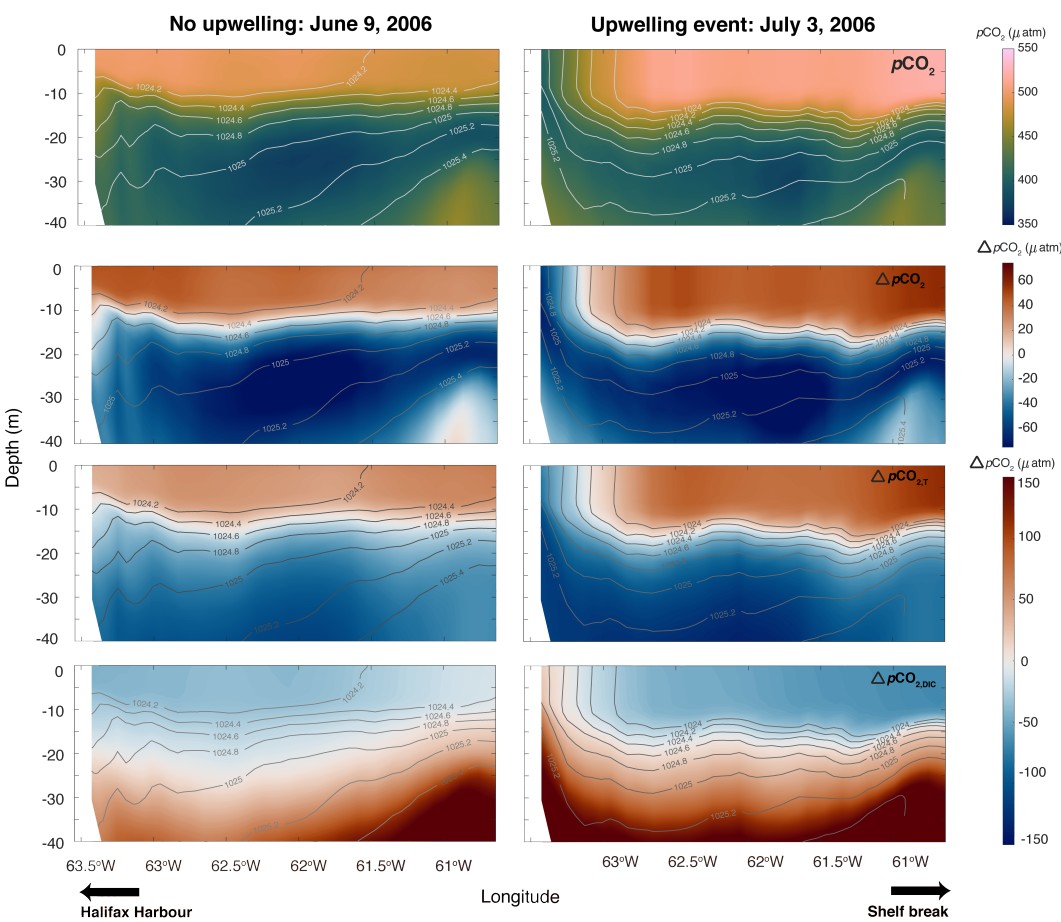

**Figure 7.** Taylor Decomposition of the upwelling event (right side; July 3, 2006) in Figure 6 compared to a non-upwelling event (left side; June 9, 2006). From top to bottom: (a) $pCO_2$, (b) overall anomaly in $pCO_2$ ($\Delta pCO_2$) from the mean $pCO_2$ in the upper 40 m, (c) anomaly in $pCO_2$ due to temperature changes ($\Delta pCO_{2,T}$), (d) anomaly in $pCO_2$ due to DIC changes ($\Delta pCO_{2,DIC}$).

of $pCO_2$, temperature and DIC with density contours along the Atlantic Condor transect for the same time slice (July 3, 2006). In these panels, the density gradients move upwards towards the coastline, consistent with upwelling events. This upwelling brings cooler temperatures and higher DIC concentrations to the surface along the coastline of Nova Scotia. The low $pCO_2$ bin ranges from $63.5^oW$ to $63^oW$ longitude in the model (approximately $63.5^oW$ to $63.3^oW$ longitude in the observations; Figure 4), and aligns with the surface area affected by the upwelling events (Figure 6) in the model. See the Supplement for more

variables along the Condor transect during the July 3, 2006, upwelling event.

Figure 7 illustrates the results of the Taylor decomposition during the July 3, 2006, upwelling event with lower $pCO_2$ nearshore compared to a snapshot without upwelling (June 9, 2006) where surface $pCO_2$ is relatively uniform. The $pCO_2$ anomalies ($\Delta pCO_2$) show the deviations in each time slice from the mean $pCO_2$ in the upper 40 m. In both time slices, the

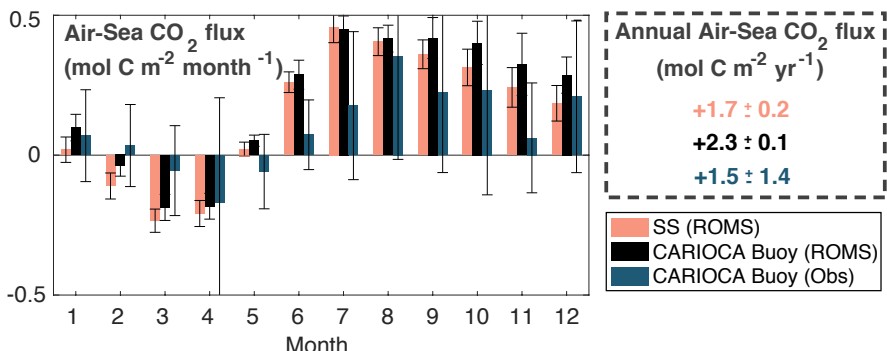

**Figure 8.** Monthly and annual air-sea $CO_2$ flux calculated from the model on the entire Scotian Shelf (pink), extracted at the CARIOCA buoy location (black), and from the buoy observations (blue). Flux is averaged over simulation years 2006-2014 for the model, and years 2007-2014 for the CARIOCA observations. Error bars are $\pm$ 1 standard deviations between years.

surface $pCO_2$ is $\sim$50 $\mu$atm higher than the mean $pCO_2$ value in the upper 40 m. However, in the upwelling case, the upwelled water is 40-50 $\mu$atm lower than the mean $pCO_2$. In both time slices, across most of the transect, temperature is acting to increase $pCO_2$ ($\Delta pCO_{2,T}$; by $\sim$50-60 $\mu$atm on June 9, 2006 and by $\sim$75-100 $\mu$atm on July 3, 2006) in the top 10-15 m from the mean value whereas DIC is acting to decrease $pCO_2$ ($\Delta pCO_{2,DIC}$; by $\sim$10-20 $\mu$atm on June 9, 2006 and by $\sim$40-50 on July 3, 2006). However, in the upwelling region on July 3, temperature has the opposite effect and is acting to decrease $pCO_2$ by $\sim$50-60 $\mu$atm and DIC is acting to increase $pCO_2$ by only $\sim$5-10 $\mu$atm from the mean $pCO_2$ in the top 40 m. The effects of alkalinity ($\Delta pCO_{2,TA}$) and salinity ($\Delta pCO_{2,S}$) are much smaller across the shelf and in both time slices (see Supplement Figure S11). Comparisons of $\Delta pCO_{2,T}$ and $\Delta pCO_{2,DIC}$ illustrate that in the upwelled region, anomalies in $pCO_2$ from temperature are larger than those from DIC. However, if water from below 30 m was upwelled, DIC would likely start to outweigh the effect of temperature on $pCO_2$.

### 4.3  Regional flux estimates

The model-simulated air-sea $CO_2$ fluxes, integrated by month and year, and averaged over the simulation from 2006-2014, for the Scotian Shelf and at the buoy location are shown in Figure 8 in comparison to the flux calculated from the CARIOCA buoy observations. The uncertainty in the model estimates is calculated as the standard deviation between years. Annually, the averaged flux between the model and observations is comparable, and the flux estimates at the buoy location are significantly larger than the shelf-wide flux estimates. The model-estimated, annually integrated flux for the Scotian Shelf shows outgassing of $CO_2$ at +1.7 $\pm$ 0.2 mol C m$^{-2}$ yr$^{-1}$. At the buoy location, just outside the upwelling region, the model estimates net outgassing of +2.3 $\pm$ 0.1 mol C m$^{-2}$ yr$^{-1}$. From the buoy observations, the annually integrated $CO_2$ flux is estimated as net outgassing at +1.5 $\pm$ 1.4 mol C m$^{-2}$ yr$^{-1}$. Although our model-derived estimate is within the upper error-bound of the observation-based estimate, it is higher, which may be due to the model's overestimation of $pCO_2$, particularly throughout the summer months. There are also some differences in the seasonal cycle. In the model, the Scotian Shelf flux is lower in magnitude than the flux

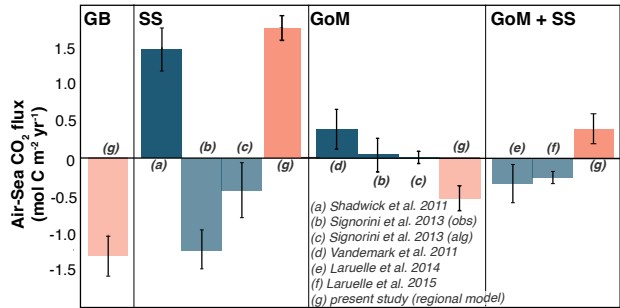

**Figure 9.** Annually integrated air-sea $CO_2$ flux for the Grand Banks (GB), Scotian Shelf (SS) and Gulf of Maine (GoM) in the model (pink) compared to literature values (blue). Positive values are net outgassing, indicated by solid bars, and negative values are net ingassing, indicated by faded bars.

at the buoy location during most of the year, and particularly from June to January. Bin 1 along the Atlantic Condor transect (Halifax Harbour/upwelling bin, Figure 1) has an annually integrated flux of +2.2 ± 0.2 mol C m$^{-2}$ yr$^{-1}$, which is comparable to the annual flux of bin 2 (Deep Panuke/shelf break bin, Figure 1) at +2.0 ± 0.2 mol C m$^{-2}$ yr$^{-1}$ and the simulated flux at the buoy location. These results indicate that cross-shelf variability in air-sea $CO_2$ fluxes is small.

Figure 9 compares the model-derived, annual flux estimates from the present study for the Scotian Shelf (+1.7 ± 0.2 mol C m$^{-2}$ yr$^{-1}$), Grand Banks (–1.3 ± 0.3 mol C m$^{-2}$ yr$^{-1}$), and Gulf of Maine (–0.5 ± 0.2 mol C m$^{-2}$ yr$^{-1}$) to previously reported estimates. The model estimate for the Scotian Shelf agrees well with the estimates from Shadwick et al. (2011) but disagrees with those from Signorini et al. (2013), Laruelle et al. (2014), and Laruelle et al. (2015). Laruelle et al. (2014) define the shelf region as a larger area that encompasses both the Scotian Shelf and Gulf of Maine. Laruelle et al. (2015) calculate one flux estimate for both the Scotian Shelf and Gulf of Maine. Signorini et al. (2013) calculates separate estimates for Gulf of Maine and Scotian Shelf. The model estimate for the Gulf of Maine agrees best with the estimates from Laruelle et al. (2014), and Laruelle et al. (2015), and disagrees with the estimates from Signorini et al. (2013), and Vandemark et al. (2011).

## 5 Discussion

We have compared the inorganic carbon dynamics in our medium complexity biogeochemical model of the northwest North Atlantic against two different observational datasets of $p$CO$_2$, one of them highly resolved in time from a CARIOCA buoy and the other with high spatial resolution along a cross-shelf transect that is occupied approximately biweekly. The largest limitation of the model is that it is unable to capture the speed and magnitude of the DIC drawdown associated with the spring bloom throughout March and April (Figure 2 and Figure 3). The simulated $p$CO$_2$ starts to decline earlier and over a longer period than in both the buoy and transect observations, and the transect shows that this timing is consistent across the whole shelf. Additionally, the model does not reach the observed $p$CO$_2$ minimum during the bloom across the whole shelf. This discrepancy appears to be a result of the bloom initiation occurring slightly too early and the bloom spanning a longer period

of time in the model, and also chlorophyll levels in the model not reaching the peak values that are observed (Figure 2a). This limitation aside, the overall seasonal cycle and switch between biological- and temperature-dominated signals in $p\text{CO}_2$ are well captured and the model simulates both the seasonal spatial and temporal variability of $p\text{CO}_2$ across the Scotian Shelf reasonably well.

Notable occurrences of spatial variability of $p\text{CO}_2$ on the Scotian Shelf occur throughout the summer months in both the model and observations. With only 1-2 clear examples of lower $p\text{CO}_2$ within $\sim$25 km of shore in the observations, we used our model to hypothesize about a possible mechanism driving this variability. In the model, we found that coastal upwelling events are driving the summertime spatial variability of $p\text{CO}_2$ on the Scotian Shelf and could explain the variability in the observations as well. The physical dynamics of coastal upwelling is well-documented on the Scotian Shelf (Petrie et al., 1987; Shan et al.,
2016). This upwelling only affects the nearshore region (within $\sim$20-40 km of shore in the model, depending on the event) where water from the cold intermediate layer is transported to the surface. In the model, this creates a coastal band of cold water at the surface that is high in DIC and low in $p\text{CO}_2$ (Figure 6). The difference between inshore and offshore temperatures ($7^o$C and $15^o$C, respectively) during these events has a larger influence on the $p\text{CO}_2$ spatial variability than the DIC variations (2050 mmol C m$^{-3}$ inshore and 2020 mmol C m$^{-3}$ offshore; Figure 6) because the thermodynamic influence of temperature
outweighs the effect of a slight increase in DIC, thus lowering $p\text{CO}_2$ (see the Taylor decomposition in Figure 7). In the example explored in the present study, the upwelled water comes from $\sim$20-25 m depth that has a $p\text{CO}_2$ approximately 100 $\mu$atm lower than the rest of the shelf. Temperature in the upwelled water is acting to lower $p\text{CO}_2$ by $\sim$150 $\mu$atm whereas DIC is acting to increase $p\text{CO}_2$ by $\sim$50 $\mu$atm compared to the rest of the shelf. If deeper water was being upwelled to the surface, DIC would likely start to be the dominant factor in setting $p\text{CO}_2$ during these events (Figure 7). For the given range of DIC values (2060
to 2020 mmol C m$^{-3}$) and a mean temperature of $11^o$C, the thermodynamic effect outweighs the effect of DIC differences for temperature changes larger than $4^o$C. Typically, it is thought that upwelling of subsurface waters rich in DIC leads to increased surface $p\text{CO}_2$ as is the case for the California Current System (CCS), encompassing the continental shelves off of Washington, Oregon, and California, where nearshore outgassing of $\text{CO}_2$ during upwelling events is well documented (Fennel et al., 2019; Chavez et al., 2017; Evans et al., 2015; Fiechter et al., 2014; Turi et al., 2014). There are, however, large differences
between the Scotian Shelf and the typical upwelling scenario of the CCS. For instance, the size and geometry of these shelves are quite different, which affects the type of water being upwelled to the surface. The California Shelf is an active margin approximately 10 km wide (Fennel et al., 2019) compared to the passive-margin Scotian Shelf with approximately 120-240 km width (Shadwick et al., 2010). As a result, the upwelling in the CCS brings DIC rich water ($\sim$2200-2250 $\mu$mol kg$^{-1}$) from deep in the water column (below 150-200 m) of the open ocean across the shelf break to the surface of the shelf (Feely et al.,
2008). On the Scotian Shelf, it is only subsurface shelf water from between $\sim$20-25 m depth that is being upwelled, which is at a similar temperature to the upwelled water in the CCS (7-8$^o$C) but at a much lower DIC concentration (2050 mmol C m$^{-3}$).

    Our regional model shows that upwelling events could be a large contributor to setting the $\text{CO}_2$ signal in the summer on the inner portion of the Scotian Shelf, acting to lower $p\text{CO}_2$ here and slightly reducing outgassing compared to the outer shelf. Throughout the remainder of the year, the $p\text{CO}_2$ distribution across the Scotian Shelf is relatively uniform (Figure
3). Comparison of the inner and outer shelf $p\text{CO}_2$ (Figure 4) shows the similar seasonality that is seen across the shelf,

both in the model results and Atlantic Condor observations. Additionally, the simulated annual air-sea $CO_2$ flux in bin 1 (upwelling bin, Figure 1) is $+2.2 \pm 0.2$ mol C m$^{-2}$ yr$^{-1}$ and similar to bin 2 (shelf break bin, Figure 1) where the annual flux is $+2.0 \pm 0.2$ mol C m$^{-2}$ yr$^{-1}$. For comparison, the annual flux for the entire shelf flux is $+1.7 \pm 0.2$ mol C m$^{-2}$ yr$^{-1}$ and the flux at the CARIOCA buoy is $+2.3 \pm 0.1$ mol C m$^{-2}$ yr$^{-1}$. Our results indicate that the short-term upwelling events in

the summer do not significantly affect the shelf-wide fluxes on an annual scale. The location of the CARIOCA buoy slightly overestimates shelfwide fluxes but is fairly representative of the shelf-wide $p$CO$_2$ dynamics overall.

     According to the model, the Scotian Shelf acts as a net source of $CO_2$ to the atmosphere ($+1.7 \pm 0.2$ mol C m$^{-2}$ yr$^{-1}$), the Gulf of Maine is a net sink of $CO_2$ ($-0.5 \pm 0.2$ mol C m$^{-2}$ yr$^{-1}$), and the Grand Banks acts as a net sink of $CO_2$ ($-1.3 \pm 0.3$ mol C m$^{-2}$ yr$^{-1}$). These results are in agreement with Shadwick and Thomas (2014) for the Scotian Shelf, and

Laruelle et al. (2014) and Laruelle et al. (2015) for the Gulf of Maine. Our results disagree, however, with results from other global (Laruelle et al., 2014) and regional studies (Laruelle et al., 2015; Signorini et al., 2013; Vandemark et al., 2011). The discrepancy in reported air-sea $CO_2$ flux between these studies is partly a result of how each study defines the area of the Scotian Shelf and Gulf of Maine. For example, Laruelle et al. (2015) calculates one estimate for both the Scotian Shelf and Gulf of Maine. The shelves of eastern North America are diverse, particularly in width and circulation features, and defining

them as a single region is not representative. Additionally, the Scotian Shelf waters are strongly influenced by cold, carbon-rich Labrador Sea water, which is not the dominant endmember south of the Gulf of Maine (Loder et al., 1998; Rutherford and Fennel, 2018; Fennel et al., 2019). Calculating a single flux estimate for the entirety of this dynamically diverse region is problematic and will yield a different estimate than when considering smaller and more specific regions. However, this only partially explains the difference in flux estimates.

Another reason is that the global SOCAT database was missing important regional data until recently. Signorini et al. (2013) used data from version 1.5 and Laruelle et al. (2014) and Laruelle et al. (2015) used data from version 2.0 of the SOCAT database. Neither of the observational datasets used in the present study were included in SOCAT versions 1.5 and 2.0. Figure 10 illustrates the difference between different SOCAT versions for seasonal $p$CO$_2$ on the Scotian Shelf. SOCAT v2020 has consistently higher average $p$CO$_2$ values than v1.5 and v2, with at least double the number of years and a much larger number

of observations going into each monthly average (on the order of 1000 to 10000 measurements in v2020 versus 100 to 1000 in v1.5 and v2). We believe that flux estimates using the updated SOCAT v2020 will agree better with our estimates and those of Shadwick and Thomas (2014) since SOCAT v2020 includes more observations with higher spatial and temporal resolution to better capture the distinct seasonal cycle here. Our study, however, only focuses on the recent seasonality of $p$CO$_2$, making it difficult to distinguish if earlier SOCAT versions miss the regional dynamics solely due to low resolution of observations,

or if the estimates from the different SOCAT versions are reflective of a shift in the behaviour of the shelf system. More work should therefore be done to better understand how variability on longer timescales could be affecting regional $p$CO$_2$ and if that variability could also be a reason for the disagreement between the different SOCAT version.

     In the present study, we have synthesized and compared our model simulations with high-resolution observations to highlight the dependence of Scotian Shelf $p$CO$_2$ seasonality on: (1) biological drawdown of DIC during the spring bloom, (2) temperature

effects throughout the summer months, and (3) wind-driven coastal upwelling events. In Figure 2d, the temperature-normalized

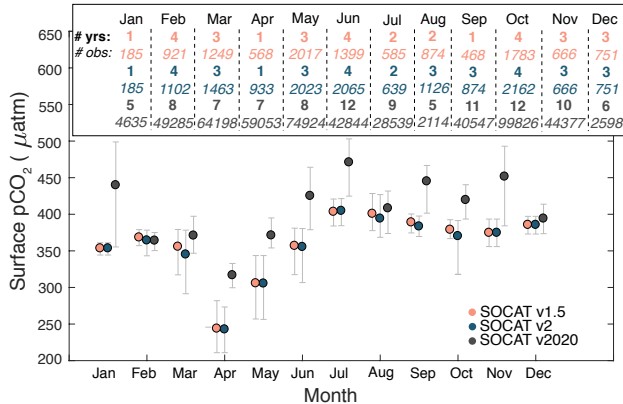

**Figure 10.** Comparison of the seasonal cycle of $pCO_2$ for the different versions of SOCAT for the Scotian Shelf, mapped to year 2006. The points indicate the mean for each month and the bars indicate the 5th and 95th percentile. Inset shows the number of years and number of observations used in each month for each version.

$pCO_2$ shows the non-thermal $pCO_2$ signal, which distinguishes the influence of biological and transport processes on $pCO_2$ (Takahashi et al., 2002). There is a clear decrease of $pCO_2$ associated with the spring bloom. The simulated decrease in $pCO_2$ is smaller than in the observations, likely due to the bloom occurring too early and over a more extended period in the model than the observations. In summer, temperature-normalized $pCO_2$ continues to decrease rather than follow the increasing temperature signal of non-normalized $pCO_2$. Previous studies have noted that, in summer, the thermodynamic signal in $pCO_2$ outweighs the influence of biological activity (Shadwick et al., 2011; Shadwick and Thomas, 2014), which could explain the differences in seasonality between $pCO_2$ and temperature-normalized $pCO_2$ in the present study. We believe this thermodynamic influence is an important factor driving the net outgassing observed on the Scotian Shelf, particularly when combined with the delivery of DIC-rich water from the Labrador Sea.

Understanding what processes presently control $CO_2$ dynamics is important for projecting how the region will be affected by changes in climate. Previous studies have suggested that the frequency and intensity of coastal upwelling could increase (e.g., Xiu et al., 2018). In the case of the Scotian Shelf, increased upwelling would lead to less outgassing or even net ingassing during summer along the coast of Nova Scotia. Climate change could therefore disproportionately affect the nearshore region here and lead to an intensification of spatial gradients. Such an upwelling signal would be in addition to the effect of increasing atmospheric $CO_2$, which may be driving the entire Scotian Shelf towards a more neutral system with less outgassing. The effect of the thermal control on Scotian Shelf $pCO_2$ is also an important aspect to consider. As temperatures continue to rise, summer $pCO_2$ values will also likely increase, potentially offsetting some of the effect of increased atmospheric $CO_2$ but also affecting production and respiration rates. Of course, none of these factors act independently and will instead combine to alter both the seasonal and spatial patterns of $pCO_2$ in the region, making the overall outcome of climate-related perturbations on the Scotian Shelf difficult to predict. However, the implementation of a regional model that resolves current conditions well, as in the present study, is an important step towards projecting future climate-related changes in the region.

## 6 Conclusions

In this study, we have validated surface $p$CO$_2$ fields on a seasonal scale from a medium-complexity regional biogeochemical model for the northwest North Atlantic shelf region against $p$CO$_2$ observations from a CARIOCA buoy and repeated cross-shelf transects from a ship of opportunity that crosses the Scotian Shelf. Except for the strength and speed of the $p$CO$_2$ drawdown associated with the spring bloom, the model simulations represent the observed spatial and temporal variability of $p$CO$_2$ on the Scotian Shelf well. Contrary to most coastal upwelling systems, upwelling events in summer are acting to lower $p$CO$_2$ within $\sim$25 km of the coastline, as cold, carbon-enriched intermediate-layer water is brought to the surface. The lowering of surface $p$CO$_2$ during these events occurs because the temperature effect leading to a lowering of $p$CO$_2$ overwhelms the increase in $p$CO$_2$ associated with DIC enrichment. We found $p$CO$_2$ to be relatively uniform across the shelf, with the exception of a narrow band impacted by summer upwelling events. Overall, the Scotian Shelf acts as a net source of CO$_2$ (+1.7 $\pm$ 0.2 mol C m$^{-2}$ yr$^{-1}$), the Gulf of Maine is a net sink of CO$_2$ (–0.5 $\pm$ 0.2 mol C m$^{-2}$ yr$^{-1}$), and Grand Banks acts as a net sink of CO$_2$ (–1.3 $\pm$ 0.3 mol C m$^{-2}$ yr$^{-1}$) in our simulation. Combination of the model simulation and the highly resolved observational datasets emphasizes that the seasonal cycle of $p$CO$_2$ is driven by strong biological drawdown of DIC in early spring and a dominant thermal control throughout the summer months. Except for the short spring bloom period, surface $p$CO$_2$ is oversaturated with respect to atmospheric values, which results in net outgassing. Ongoing changes in climate and carbon cycling will likely alter both the seasonal and spatial patterns of $p$CO$_2$ on the Scotian Shelf.

*Code and data availability.* The ROMS model code can be accessed at http://www.myroms.com (last access: 1 May 2011). Here version 3.5 was used. The Atlantic Condor observations can be found on the SOCAT database at https://www.socat.info/

*Author contributions.* KR and KF conceived the research questions of this study. KR carried out the model simulations and analyses. DA, DW and HT implemented observational platforms and contributed the subsequent datasets. KR and KF discussed the results and wrote the manuscript, with inputs from DA, DW and HT.

*Competing interests.* The authors have no competing interests.

*Acknowledgements.* We acknowledge funding by the Marine Environmental Observation, Prediction and Response Network (MEOPAR). This work was additionally funded in part by the Canada Excellence Research Chair (CERC) in Ocean Science and Technology at Dalhousie University and Canada Foundation for Innovation (CFI) project number 29011. We would like to thank the captain and crew of Atlantic Condor for their continuing support in operating the Dal-SOOP underway system. Mike Vining, Jeremy Lai, Dan Kehoe, Kitty Kam and Jordan Sawler contributed to the design, installation and maintenance of the underway system. We also are grateful for the use of observational

datasets to initialize our model from Kumiko Azetsu-Scott (Department of Fisheries and Oceans Canada) and Alfonso Mucci (McGill University). We would also like to acknowledge the use of the scientific colourmaps lapaz, vikO, and batlow (Crameri, 2018) used in this study. We are additionally grateful for the three reviewers of our paper and their constructive feedback.

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
