# Peer review of "A modeling study of temporal and spatial $pCO_2$ variability on the biologically active and temperature-dominated Scotian Shelf"

_Biogeosciences, 2021_

## Author Comment (AC1)

**A modeling study of temporal and spatial pCO2 variability on the biologically active and temperature-dominated Scotian Shelf**

**Response to Comments by Reviewer 1**

(Reviews are included in black font; Responses are in blue font)

This work seeks to identify the role of local event-scale variability – namely upwelling – in determining the regional air-sea carbon dioxide fluxes over the Scotian Shelf through the integration of several different data sets as well as the use of a regional numerical model. The paper features wonderful contextualization of previous flux estimates with observational limitations and integration of multiple kinds of data for this regional problem. The problem itself is quite timely as recent work has identified that the coastal ocean rates of change in carbon dioxide may always reflect the global changes. The manuscript requires additional details in the methods section – most notably about the regressions used to drive the initial and boundary conditions and river values, some issues with time surrounding the observations used and the simulation years, as well as the methods pertaining to evaluation of the model itself. In additional, more attention needs to be paid to the role of the Revelle Factor in driving these interregional differences between the upwelling on this shelf and the CCS. Finally – and most importantly – the authors need to clarify how the upwelling event contributes to the shelf wide estimates more clearly. The paper would be publishable in Biogeosciences if these issues can be addressed by the author team. More specific comments follow.

**Response:** We appreciate the constructive comments and will pay close attention in our revision to clarify the objectives of our study and to provide or emphasize the methodological details requested by the reviewer. The main objectives of our study are to show (1) that the Scotian Shelf, as a whole, acts as a net source of  $CO_2$  to the atmosphere, (2) that local processes drive seasonal and spatial variability of pCO2, and (3) to present an assessment of how well our regional model captures these processes. The methodological details will be provided as detailed in the responses below. A discussion of the Revelle factor is beyond the intended scope of this study. Likewise, as explained in more detail below, the upwelling event is not a major contributor to the shelf-wide air-sea flux and we did not mean to suggest it is. This will be stated clearly in the manuscript (see response to next comment).

**Major Comments:**

The main message appears to be that local processes are important for carbon content of the temperate Scotian shelf region. In the context of that message, the authors need to show how the localized upwelling event contributed to the overall regional flux somehow. One way might be to show this flux as a map. While there is quite a bit of information on the in situ observed location's variability, there is very little about how that compares to the region as a whole – is it representative? For instance, where in Figure 1 does this upwelling occur (at the buoy and along the black line/transect?) – and how does the simulated flux at the surface of the entire region compare to this localized event? How fine of a resolution do we need to observe to get the shelf-based flux estimate direction right? Also, how does this flux compare with other regional/broader scale fluxes reported for the North Atlantic?

**Response:**

First, we would like to emphasize again that the main objectives of our study are to show (1) that the Scotian Shelf, as a whole, acts as a net source of  $CO_2$  to the atmosphere, (2) that local processes drive seasonal and spatial variability of  $pCO_2$ , and (3) to present an assessment of how well our regional model captures these processes. We accomplish this by combining two high-resolution data sets, a timeseries (CARIOCA buoy) and regular cross-shelf transects (Atlantic Condor cruises) with a high-resolution regional model.

**In response to:** "the authors need to show how the localized upwelling event contributed to the overall regional flux somehow.":**

One of the main messages of our paper is that the flux across the shelf is relatively uniform and that these localized summer upwelling events do not contribute significantly to shelf-wide fluxes but would be more important locally. We believe this is an interesting contrast to other shelves with summer upwelling (e.g. the California Current System or CCS), where these events have been shown to contribute significantly to air-sea fluxes. This will be stated more clearly in section 4.3 Regional Flux Estimates (line 237) where we will add:

"Bin 1 along the Atlantic Condor transect (upwelling bin, Figure 1) has an annually integrated flux of 2.3 mmol C  $m^{-2}$  yr-1 compared to bin 2 (shelfbreak bin, Figure 1) with an annual flux of 2.2 mmol C  $m^{-2}$  yr-1."

In section 5 Discussion (line 284) we will add:

"Additionally, annual air-sea  $CO_2$  fluxes in bin 1 (upwelling bin, Figure 1) are estimated in the model to be 2.3 mmol C m-2 yr-1 compared to bin 2 (shelfbreak bin, Figure 1) with an annual flux of 2.2 mmol C m-2 yr-1. This is compared to the entire shelf flux of 1.9 mmol C m-2 yr-1 and the flux at the CARIOCA buoy of 2.4 mmol C m-2 yr-1. Our results indicate that the upwelling events are such short term events that they do not significantly affect the fluxes and that shelfwide fluxes on an annual scale are relatively uniform."

**In response to: "where in Figure 1 does this upwelling occur (at the buoy and along the black line/transect?)":**

We will add the location of the CARIOCA buoy and condor transect to the upwelling figure (top panel Figure 6).

**In response to: "How fine of a resolution do we need to observe to get the shelf-based flux estimate direction right?":**

As mentioned above, air-sea flux is rather homogenous across the shelf and the localized upwelling events do not noticeably affect shelf-wide air-sea flux. Although more observations would be better, of course, it appears that the combination of the high-resolution time series and the cross-shelf transects provide adequate resolution to support our conclusions.

**In response to: "how does this flux compare with other regional/broader scale fluxes reported for the North Atlantic":**

We provide such reported fluxes in Figure 9 where we compare our flux estimates to other regional and global fluxes reported for the region (from Grand Banks to Gulf of Maine).

Secondly, it is critical to clarify time in this work. 2005 was the year when the warming started intensely on the east coast of North America. The model runs happen before that, but the comparisons are to data after that.... How does that impact the results? What about the time variability of carbon dioxide in the atmosphere over these various intervals? **Response:** We will update our manuscript using an extended simulation from 1999-2014. This would encompass the observation years from the CARIOCA buoy (2007-2014). When plotting model and observations from different years in the same graph, we will perform a simple detrending where we map these observations onto the common year using the long-term atmospheric pCO2 trend of ~ + 2  $\mu$ atm/year. Analysis of the warming observed on the east coast of North America is outside of the intended scope of this paper, which aims to address seasonal variability. A model analysis of long-term trends with the same model is forthcoming.

The comparison to the California Current or other traditionally upwelling situations is not entirely accurate as the vertical gradient in DIC (presented in the figure here) is nearly half what it is in the CCS (Feely et al. 2004). The phytoplankton growth at the surface is quite efficient unless the winds blow too strongly and the phytoplankton can no longer grown in place. This aspect of the upwelling system is neglected in the text. The signature of the phytoplankton drawdown can be seen very far offshore as it takes nearly a year for CO2 to equilibrate at the surface. In addition, the two systems likely experience very different temperature, salinity, and alkalinity parameter spaces – all of which are important to consider for the response of the carbon system.

**Response:**

We fully agree with the Reviewer that the Scotian Shelf and the CCS behave very differently. In fact, we believe this is one of the interesting findings of our study and we would like to make this clearer in the Discussion section by referring and comparing to Feely et al. (2008). More specifically, we will emphasize how these two systems are different, bringing in reference to the transects in Figure 2 of Feely et al. (2008) (included below) and compare to our Figure 6. We will point out the differences in size and shape of the two shelf regions, which lead to different types of water being upwelled. In the CCS, the shelf is much narrower than the Scotian Shelf (California shelf ~10 km; Scotian Shelf ~120-240 km), which means that upwelling in the CCS brings water from deeper in the water column (below 150 - 200 m) of the open ocean across the shelf break to the surface. In contrast, upwelling on the Scotian Shelf brings intermediate-layer water (from between 20 – 100 m) from the shelf itself. While the temperatures of the upwelled water are similar in both systems (~8°C), upwelled water in the CCS has much higher DIC concentrations (~2200-2250 umol kg-1) than the Scotian Shelf (2060 mmol m-3). This will be added to the Discussion section.

The Revelle Factor influence on the differences between what is observed on the Scotian Shelf and in the CCS should be included – for an example described in more detail see here: https://www.sciencedirect.com/science/article/pii/S0278434317303643#f000 Because the Revelle Factor is important to consider within the context of this issue, it would be important to evaluate DIC and TA with in situ observations locally, here within this manuscript. Please add DIC and TA evaluation of the model fields. Do observations of these fields exist for the simulated period?

**Response:**

We agree with the Reviewer that an evaluation of DIC and TA will be useful. We will add this using in-situ observations available from the Department of Fisheries and Oceans (DFO) Atlantic Zone Monitoring Program (AZMP). We will focus on DIC and TA. A discussion of the Revelle factor is beyond the intended scope of this manuscript.

The methods require quite a bit more detail. Specifically, what is the model skillful in (Lines 112) from other studies? Was it evaluated mostly at the surface? Over annual timescales? Or events like in this work? The K1 and K2 constants chosen are not meant for regions that

experience a lot of freshwater influence. Can you justify their choice in this region by discussing the salinity ranges that this region observes? What atmospheric carbon dioxide concentration was used?

**Response:**

**Regarding Lines 112ff**, we would like to expand the text as follows:**

"For a detailed description and validation of the biological model, we refer to Laurent et al. (2020), who showed that it outperforms global models for the region in terms of model skill at representing nitrate and chlorophyll. Our model was evaluated on a se

---

## Author Comment (AC2)

**A modeling study of temporal and spatial pCO2 variability on the biologically active and temperature-dominated Scotian Shelf**

**Response to Comments by Reviewer 2**
(Reviews are included in black font; Responses are in blue font)

The authors use output from a regional oceanic biogeochemical model and mooring/ship-board observations to study the seasonal cycle of surface pCO2 and sea-air CO2 fluxes. The general findings are that the Scotian Shelf acts as a net annual source of CO2 to the atmosphere and that biological activity and temperature are the main drivers of the pCO2 variability. The authors also try to show that coastal upwelling is responsible for low near-shore surface pCO2 in summer. Overall, I find the manuscript well organized. However, I have several concerns (see below) that need to be addressed.
**Response:** We appreciate the constructive comments and propose to address them as described below.

Line 14: Might be good to mention here by how much pCO2 changes due to this steep increase in temperature.
**Response:** Agree, we will modify the text as follows (addition in bold italics): "*Surface pCO$_2$ undergoes a strong seasonal cycle **with an amplitude of ~200-250 µatm**. **These changes are** associated with both a strong biological drawdown of Dissolved Inorganic Carbon (DIC) in spring **(corresponding to a decrease in pCO$_2$ of 100-200 µatm)**, and pronounced effects of temperature, which ranges from 0ºC in the winter to near 20ºC in the summer**, resulting in an increase in pCO$_2$ of ~ 200-250 µatm.***"

Lines 33-36: Since you specify the type of measurements that suggest that the Scotian Shelf is a net CO2 source, it would be interesting to know what type of data suggests that it is a net sink.
**Response:** It is the same type of measurement (surface ocean pCO$_2$), but different data sets are used in these studies. We propose to clarify this by stating explicitly that these other studies use the SOCAT database.

Line 63: I would be careful calling any model "accurate"! If the model has been evaluated properly (if the region has an adequate amount of observations), then I bet these studies identified some deficiencies. I would suggest to briefly summarizing the previous model evaluation here and state unknowns due to lack of data, if applicable.
**Response:** Agree, we will modify this as follows (added/modified text in bold italics):
 "*In the present study, we employ a high-resolution biogeochemical model of the northwest North Atlantic to examine the magnitude, variability and sign of the air-sea CO$_2$ flux on the Scotian Shelf. Previous studies **evaluated** our **model's ability to** represent the physical (Brennan et al. 2016, Rutherford & Fennel 2018) and biological (Laurent et al. 2020) dynamics of the region.*"
We will also add more detail about these evaluations in the methods, where we describe the physical and biological model set up.

Line 105: What are the possible implications of using a river climatology to force the model? Is anything known about interannual or longterm changes to the riverine input?

**Response:** The intended purpose of this paper is to focus on the seasonal variability not interannual or long-term changes. River inputs make up a very small fraction of the water on the Scotian Shelf (see Rutherford & Fennel 2018) and variations in riverine chemistry over this short period would be negligible. We will emphasize in the revised manuscript that the focus of this study is on a seasonal scale.

Line 102: Again, a brief summary of deficiencies and skills of the model would be good.
**Response:** Agree, we will add the following (addition/modification in bold italics):

 "*For a detailed description and validation of the biological model, we refer to Laurent et al. (2021),* **who compared the model output with glider transects of temperature, salinity and chlorophyll, and in situ measurements of chlorophyll and nitrate. They showed that the model outperforms global models in this region for all variables and that the timing of the spring bloom is well represented, but the model slightly underestimates the magnitude of the bloom and tends to slightly overestimate nitrate throughout the year.** "

 "*Full details on the physical model setup and its validation can be found in Brennan et al. (2016) and Rutherford & Fennel (2018).* **These studies have shown that our model simulates the vertical structure and seasonal cycling of temperature and salinity on the shelf well. The model captures mesoscale features and coastal upwelling events, and simulates the volume transport throughout the region in agreement with observation-based estimates.**"

Line123: Why is it drifting and how does the nudging impact the actual model skill. I was surprised that so much nudging was done for a relatively small model domain. Are the nudged areas not used in the analysis? And if these areas are used, how do you deal with them? Would be helpful to show the nudged areas in Figure 1.
**Response:** The nudging zones along the boundary are not used in the analysis, which we will state explicitly in the revised manuscript. It is common to apply boundary nudging in regional domains. The nudging timescale is long (60 days at the boundary, linearly decreasing to zero at the inner edge of the nudging zone). This means that nudging is weak. Since the internal dynamics of the Labrador Sea, which set the seasonal cycle of physical and biogeochemical conditions at the northeastern boundary, are not represented in our regional shelf-focussed domain, it is not surprising that conditions would be drifting slowly without nudging. Nudging benefits model skill by eliminating unrealistic drifts.

Line 131: Model spin-up of a biogeochemical model usually takes 6-10 years. Can you show that 1 year is enough and the model won't drift anymore? For example run the model for 10 years perpetually, using the 2000 conditions. Does DIC remain relatively stable, without drifting?
**Response:** We have now run a longer simulation (1999-2014), which we will analyze to replace the current 6-year simulation (1999-2004) presented in the manuscript. We will focus on the later years of this simulation and can include a figure in the supplement with a timeseries of DIC to illustrate the spin-up period.

Line 147: Need to label the location of the Halifax and Deep Panuke gas platform.
**Response:** Agree, we can add this.

Line 163: "from top to bottom…" belongs into caption and not into main text. Also, describe method you used to temperature normalize pCO2 in caption.
**Response:** Agree, we will modify this.

Line 164: To me it is confusing to talk about days and months. I would just stick to months, since days are less obvious – The reader would have to first convert to the month before understanding what time of the year you are referring to. I don't see how pCo2 is relatively constant between day 0 to 75. Are you referring to the temp normalized pCO2? But even temp normalized pCO2 is increasing during this time. Might be better to give a range here?
**Response:** We prefer to use the day of year for this section because it is more specific than referring to months and believe it is clear.

Line178: add "buoy" to "… at the low end of the **buoy** observations
**Response:** Agree.

Line 182: I don't think the word "consistent" is appropriate here? The model seems to underestimate the DIC drawdown due to primary production compared to both types of observations (temp norm. pCO2).
**Response:** Agree. We will modify this line to "The bloom-related minimum in $pCO_2$ in the model is approximately 50-75 µatm higher than the buoy observations and approximately 25-50 µatm higher than the Atlantic Condor observations."

Line 189: verb is missing.
**Response:** We will correct this.

Line 209: Figure 4 shows how the model struggles to simulate the spatial variability, which should be pointed out.
**Response:** Agree, we will add a sentence at the end of the first paragraph stating that the model does not show the small scale variations in $pCO_2$ that are seen in the Condor transect. However, we would like to add that this is not surprising. It is common that models produce much less variations than underway $pCO_2$ observations. We would also point out that underway measurements are prone to many errors and that the variations may at least partly be due to measurement artifacts.

Line 212: add east or west to longitude description
**Response:** Yes, we will add this throughout the paper.

Line 216: I don't think these events are all that obvious in the observations. There were only a total 3 inner shelf observations during this time period, two of which are actually higher than an outer shelf observation point (also the only one during this period). I agree, that this is obvious in the model, but would be more careful with this statement for the observations. I just don't think that the observations can be interpreted that way… Im also not convied by the proposed mechanism that leads to low pCO2, despite high DIC. What does the salinity profile look like? I think this section needs something like a Taylor decomposition to show that what is responsible for the low pCO2 (see details in
Rheuban, J. E., Doney, S. C., McCorkle, D. C., and Jakuba, R. W.: Quantifying the effects of nutrient enrichment and freshwater mixing on coastal ocean acidification, J. Geophys. Res.-Oceans, 124, 9085–9100, https://doi.org/10.1029/2019JC015556, 2019.

Or

Hauri, C., Schultz, C., Hedstrom, K., Danielson, S., Irving, B., Doney, S.C., Dussin, R., Curchitser, E.N., Hill, D.F, and Stock, C.A.: A regional hindcast model simulating ecosystem dynamics, inorganic carbon chemistry, and ocean acidification in the Gulf of Alaska, Biogeosciences, 17, 3837–3857, https://doi.org/10.5194/bg-17-3837-2020, 2020.

**Response:** The observations with higher $pCO_2$ nearshore have been removed from the analysis, due to sensor error. There is 1 instance in the observations where this low $pCO_2$ signal is very clear, which is associated with cold temperatures (refer to Figure 5). We agree that with only 1 instance in the observations of low $pCO_2$ nearshore, it is difficult to draw unequivocal conclusions about the mechanisms driving this. We will therefore rephrase this section as using the model to hypothesize what could be driving this low $pCO_2$ nearshore.

We additionally plan to add more variables along the transect to the supplementary info. We also really like the idea of doing a Taylor Decomposition and plan to add this as well.

Line 264: "Accurate" means: "correct in all details; exact"- as menti0oned earlier, I yet have to see a model that can be described as "accurate". I would tone it down… especially because you start the sentence with "This limitation aside…"
**Response:** Agreed, will remove the word "accurately."

Line 270: would be nice to calculate how much the temperature change affects pCO2 and how much DIC increases affect pCO2….
**Response:** Agreed, we think adding the Taylor Decomposition will accomplish this.

**Figures – I really like the color choices of the figures!**
**Response:** Thank you!

Figure 1: It would be nice to give the reader a better understanding of where the Scotian Shelf is located. Maybe a zoomed-out map as an insert? Label all location names you are mentioning in the paper e.g. Halifax Harbor. What are bin 1 and bin 2? Please describe in caption. Also, LAt and Lon labels are missing, including whether it is north or south, and east or west. This should be adjusted for all figures throughout.
**Response:** We will make these changes.

Figure 2: Correct "Glider obser**v**ations"
Identify grey band in legend for consistency.
What are the two different x-axis?
**Response:** We can make these changes. We included a DOY and month x-axis to help the reader (since we refer to DOY in the text).

Figure 4: What are these inserts? Zoom in? Does not seem to show what you see in the smaller box below. This figure is kind of confusing. What are we actually looking at? Are there 365/5 lines total per figure?
**Response:** We will make the caption for this figure clearer and more descriptive.

Figure 5: On the left, there is no top and lower panel... please adjust accordingly. Also, maybe identify "thick black line" as "vertical black line"
**Response:** Agree, these have been modified.

Figure 6:Please identify the variable that goes with each unit next to the colorbar. Always good to specify units of all variables in the caption too. Also, define abbreviations in all figures e.g. dissolved inorganic carbon (DIC). Figures and captions should be readable without reading the manuscript. Since you refer from figure 5 to this figure, you should mention here that July 11, 2000 is indicated in figure 5. Please show the transect line again in the map and label it with lat and lon. Why not also show a profile of pCO2 here to make the point that pCO2 decreases during upwelling event.
**Response:** Agree. We will make these changes.

Figure 7: Add "the" to …the values from **the** nearshore bin…
**Response:** Done.

Figure 8: What do the error bars mean? What are they based on? What are the numbers behind +/-? 1 STD? Needs to be defined in caption.
**Response:** Error bars are standard deviation between years. We will add this to the figure caption.

Figure 9: Why are some bars faded? Are all ingassing bars faded? Needs to be defined.
**Response:** Yes, all ingassing bars are faded. We will add this to the figure caption.

---

## Author Response (AR1)

**A modeling study of temporal and spatial pCO$_2$ variability on the biologically active and temperature-dominated Scotian Shelf**

**Response to Comments by Reviewer 1**
(Reviews are included in black font; Responses are in blue font)

This work seeks to identify the role of local event-scale variability – namely upwelling – in determining the regional air-sea carbon dioxide fluxes over the Scotian Shelf through the integration of several different data sets as well as the use of a regional numerical model. The paper features wonderful contextualization of previous flux estimates with observational limitations and integration of multiple kinds of data for this regional problem. The problem itself is quite timely as recent work has identified that the coastal ocean rates of change in carbon dioxide may always reflect the global changes. The manuscript requires additional details in the methods section – most notably about the regressions used to drive the initial and boundary conditions and river values, some issues with time surrounding the observations used and the simulation years, as well as the methods pertaining to evaluation of the model itself. In additional, more attention needs to be paid to the role of the Revelle Factor in driving these interregional differences between the upwelling on this shelf and the CCS. Finally – and most importantly – the authors need to clarify how the upwelling event contributes to the shelf wide estimates more clearly. The paper would be publishable in Biogeosciences if these issues can be addressed by the author team. More specific comments follow.

**Response:** We appreciate the constructive comments and have paid close attention in our revision to clarify the objectives of our study and to provide or emphasize the methodological details requested by the reviewer. The main objectives of our study are to show (1) that the Scotian Shelf, as a whole, acts as a net source of CO$_2$ to the atmosphere, (2) that local processes drive seasonal and spatial variability of pCO$_2$, and (3) to present an assessment of how well our regional model captures these processes. The methodological details are provided as detailed in the responses below. A discussion of the Revelle factor is beyond the intended scope of this study. Likewise, as explained in more detail below, the upwelling event is not a major contributor to the shelf-wide air-sea flux and we did not mean to suggest it is. This is now stated clearly in the manuscript (see response to next comment).

Major Comments:

The main message appears to be that local processes are important for carbon content of the temperate Scotian shelf region. In the context of that message, the authors need to show how the localized upwelling event contributed to the overall regional flux somehow. One way might be to show this flux as a map. While there is quite a bit of information on the in situ observed location's variability, there is very little about how that compares to the region as a whole – is it representative? For instance, where in Figure 1 does this upwelling occur (at the buoy and along the black line/transect?) – and how does the simulated flux at the surface of the entire region compare to this localized event? How fine of a resolution do we need to observe to get the shelf-based flux estimate direction right? Also, how does this flux compare with other regional/broader scale fluxes reported for the North Atlantic?

**Response:**

First, we would like to emphasize again that the main objectives of our study are to show (1) that the Scotian Shelf, as a whole, acts as a net source of $CO_2$ to the atmosphere, (2) that local processes drive seasonal and spatial variability of $pCO_2$, and (3) to present an assessment of how well our regional model captures these processes. We accomplish this by combining two high-resolution data sets, a timeseries (CARIOCA buoy) and regular cross-shelf transects (Atlantic Condor cruises) with a high-resolution regional model.

**In response to: "the authors need to show how the localized upwelling event contributed to the overall regional flux somehow.":**

One of the main messages of our paper is that the flux across the shelf is relatively uniform and that these localized summer upwelling events do not contribute significantly to shelf-wide fluxes but would be more important locally. We believe this is an interesting contrast to other shelves with summer upwelling (e.g. the California Current System or CCS), where these events have been shown to contribute significantly to air-sea fluxes.

This is now stated more clearly in section 4.3 Regional Flux Estimates (line 287ff in the revised manuscript) where we have added:

"*Bin 1 along the Atlantic Condor transect (Halifax Harbour/upwelling bin, Figure 1) has an annually integrated flux of +2.2 ± 0.2 mmol C m$^{-2}$ yr$^{-1}$, which is comparable to the annual flux of bin 2 (Deep Panuke/shelfbreak bin, Figure 1) at +2.0 ± 0.2 mmol C m$^{-2}$ yr$^{-1}$ and the simulated flux at the buoy location. These results indicate that cross-shelf variability in the air-sea $CO_2$ flux is small.*"

In section 5 Discussion (line 344ff in the revised manuscript) we have added:

"*Additionally, the simulated annual air-sea $CO_2$ flux in bin 1 (upwelling bin, Figure 1) is +2.2 ± 0.2 mmol C m$^{-2}$ yr$^{-1}$ and similar to bin 2 (shelfbreak bin, Figure 1) where the annual flux is +2.0 ± 0.2 mmol C m$^{-2}$ yr$^{-1}$. For comparison, the annual flux for the entire shelf flux is +1.7 ± 0.2 mmol C m$^{-2}$ yr$^{-1}$ and the flux at the CARIOCA buoy is +2.3 ± 0.1 mmol C m$^{-2}$ yr$^{-1}$. Our results indicate that the short-term upwelling events in summer do not significantly affect the shelfwide fluxes on an annual scale.*"

**In response to: "where in Figure 1 does this upwelling occur (at the buoy and along the black line/transect?)":**

We have added the location of the CARIOCA buoy and condor transect to the upwelling figure (top panel Figure 6).

**In response to: "How fine of a resolution do we need to observe to get the shelf-based flux estimate direction right?":**

As mentioned above, air-sea flux is rather homogenous across the shelf and the localized upwelling events do not noticeably affect shelf-wide air-sea flux. Although more observations would be better, of course, it appears that the combination of the high-resolution time series and the cross-shelf transects provide adequate resolution to support our conclusions.

**In response to: "how does this flux compare with other regional/broader scale fluxes reported for the North Atlantic":**

We provide such reported fluxes in Figure 9 where we compare our flux estimates to other regional and global fluxes reported for the region (from Grand Banks to Gulf of Maine).

Secondly, it is critical to clarify time in this work. 2005 was the year when the warming started intensely on the east coast of North America. The model runs happen before that, but the comparisons are to data after that…. How does that impact the results? What about the time variability of carbon dioxide in the atmosphere over these various intervals?

**Response:** We have updated our manuscript using an extended simulation from 1999-2014. We now focus on model years 2006-2014, which encompasses the observation years from the CARIOCA buoy (2007-2014). When plotting model and observations from different years in the same graph, we perform a simple detrending where we map these observations onto the common year using the long-term atmospheric $pCO_2$ trend of $\sim + 2$ µatm/year (see lines 190ff and Supplement Figure S1). Analysis of the warming observed on the east coast of North America is beyond the intended scope of this paper, which aims to address seasonal variability (now emphasized throughout the revised manuscript, for example at lines 11, 67, 68, 310, 371, 396). A model analysis of long-term trends with the same model is forthcoming.

The comparison to the California Current or other traditionally upwelling situations is not entirely accurate as the vertical gradient in DIC (presented in the figure here) is nearly half what it is in the CCS (Feely et al. 2004). The phytoplankton growth at the surface is quite efficient unless the winds blow too strongly and the phytoplankton can no longer grown in place. This aspect of the upwelling system is neglected in the text. The signature of the phytoplankton drawdown can be seen very far offshore as it takes nearly a year for CO2 to equilibrate at the surface. In addition, the two systems likely experience very different temperature, salinity, and alkalinity parameter spaces – all of which are important to consider for the response of the carbon system.

**Response:**

We fully agree with the Reviewer that the Scotian Shelf and the CCS behave very differently. In fact, we believe this is one of the interesting findings of our study and we have made this clearer in the Discussion section by referring and comparing to Feely et al. (2008).

We have added the following to the discussion section (see lines 328ff in the revised manuscript): "*There are, however, large differences between the Scotian Shelf and the typical upwelling scenario of the CCS. For instance, the size and geometry of these shelves are quite different, which affects the type of water being upwelled to the surface. The California Shelf is an active margin shelf approximately 10 km wide (Fennel et al. 2019) compared to the passive-margin Scotian Shelf with approximately 120-240 km width (Shadwick et al. 2010). As a result, upwelling in the CCS brings DIC rich water ($\sim$2200-2250 umol kg$^{-1}$) from deep in the water column (below (150-200 m) of the open ocean across the shelf break to the surface of the shelf (Feely et al. 2008). On the Scotian Shelf, it is only subsurface shelf water from between $\sim$20-25 m depth that is being upwelled, which is at a similar temperature to the upwelled water in the CCS (7-8$^o$C) but at a much lower DIC concentration (2050 mmol C m$^{-3}$).*"

The Revelle Factor influence on the differences between what is observed on the Scotian Shelf and in the CCS should be included – for an example described in more detail see here: https://www.sciencedirect.com/science/article/pii/S0278434317303643#f000

Because the Revelle Factor is important to consider within the context of this issue, it would be important to evaluate DIC and TA with in situ observations locally, here within this manuscript. Please add DIC and TA evaluation of the model fields. Do observations of these fields exist for the simulated period?

**Response:**

We agree with the Reviewer that an evaluation of DIC and TA is useful. We have added this using in-situ observations available from the Department of Fisheries and Oceans (DFO) Atlantic Zone Monitoring Program (AZMP) and have added a brief evaluation of DIC and TA to these observations in the supplement (Section S4; Figures S6 – S8). A discussion of the Revelle factor is beyond the intended scope of this manuscript.

The methods require quite a bit more detail. Specifically, what is the model skillful in (Lines 112) from other studies? Was it evaluated mostly at the surface? Over annual timescales? Or events like in this work? The K1 and K2 constants chosen are not meant for regions that experience a lot of freshwater influence. Can you justify their choice in this region by discussing the salinity ranges that this region observes? What atmospheric carbon dioxide concentration was used?

**Response:**

Regarding Lines 114ff, we have expanded the text as follows (Lines 114-119 in revised manuscript; additions in bold italics): "*For a detailed description and validation of the biological model, we refer to Laurent et al. (2021), **who compared the model output with glider transects of temperature, salinity and chlorophyll, and in situ measurements of chlorophyll and nitrate. The model was evaluated on a seasonal scale for the entire model domain, mainly in the surface (top 100 m). Laurent et al. (2021) showed that the model outperforms global models for the region for all variables and that the timing of the spring bloom is well represented, but the model slightly underestimates the magnitude of the bloom and tends to slightly overestimate nitrate throughout the year.***"

Regarding our use of our K1 and K2 constants, we have modified the text to (lines 120-123 in the revised manuscript; additions in bold italics): *"...we use dissociation constants (K1 and K2) from Millero et al. (1995) using Mehrbach et al. (1973) data on the seawater scale **which are deemed appropriate for the typical salinity ranges from 27 to 36.6 in the model domain (lower salinities are highly localized in the Gulf of St. Lawrence Estuary)**."*

Regarding atmospheric $CO_2$ concentrations:

As stated on lines 145-146 in the original text (lines 123 in revised manuscript): "*Atmospheric $pCO_2$ is set to the seasonal cycle and secular trend derived from Sable Island monitoring data contributed by Environment Canada's Greenhouse Gas Measurement Program (Environment and Climate Change Canada, 2017).*"

We have added the trend equation with seasonal cycle to the Supplement with a figure illustrating (see Supplement S2 and Figure S1).

Most importantly in the methods – the boundary condition DIC and TA relationships and river concentrations require additional documentation. In the case of the boundary conditions, they appear to rely solely on data from the winter months from an unspecified location. Can you add these relationships to supplemental? And describe the data that they rely on? Are they from a

similar time period that was simulated? Were adjustments made for time in the DIC field if they were observed more than 5 years earlier/later than the simulations? There are existing hydrographic relationships in the region and globally that could be used instead (McGarry et al. 2021; Xu et al. 2020; CANYON; LIAR) – why generate a new one?

**Response:**
We have added a more detailed description of the DIC and TA data in the Methods section (they are from DFO's AZMP program mentioned above). The relationships are reported there already (lines 132-138) we have modified the text as follows:

Lines 137ff in revised text (updated text in bold italics): "*The model is initialized on January 1, 1999 from Urrego-Blanco and Sheng's (2012) solution for temperature and salinity. Nitrate ($NO_3^-$) concentrations are initialized from regional climatologies as in Laurent et al. (2020).* **DIC and TA initial and boundary conditions were created from observationally based relationships with temperature (T) and salinity (S) using bottle data from regional cruises from 1997-2011 encompassing as far south as the Gulf of Maine and as far north as the Labrador Sea (observations from DFO's AZMP program, see: dfo-mpo.gc.ca/science/data-donnees/azmp-pmza/index-eng.html#publications). Initialization relationships used only observations from December, January and February (TA = 43S + 800, $r^2$ = 0.96; DIC = 1153 – 21.6T + 29.1S – 0.41$T^2$ + 0.63ST, $r^2$ = 0.90). Boundary conditions used observations that encompass the entire year (TA = 41S + 875, $r^2$ = 0.92; DIC = 912.6 – 2.4T + 35.7S – 0.45$T^2$ + 0.12ST, $r^2$ = 0.80).***"

Why did we not use other relationships? Aside from the obvious reason of timing (we have been working on the model for a few years while McGarry et al. 2021 and Xu et al. 2020 were just recently published), we know it is crucial to use observations from our shelf region. McGarry et al. (2021) focuses on Gulf of Maine and does not include most of our study region. Similarly, Xu et al. 2020 focuses on MAB and SAB, and not our study region. Since the Gulf of Maine, MAB and SAB are all more strongly influenced by Gulf Stream water than the upstream shelves that we focus on, it is important to use hydrographic relationships that are specific to our region of focus.  We do not believe the CANYON fields are appropriate as they are derived from open-ocean not shelf data. Furthermore, as CANYON requires the use of oxygen data in addition to temperature and salinity, which we do not have access to for the entire region, and any workarounds would introduce errors. Likewise, we did not use LIAR because it was optimized for the open ocean not the shelf.

Finally, the point that the upwelling event signal leads to reduced outgassing compared to the rest of the shelf (Line 280-281) is not clearly shown and is related to the main point of the work. The reader is still considering (because none of the other fields were shown) that maybe the phytoplankton growth rate in relationship to the winds -documented in Evans et al. (2015) could also be contributing to this. What does the subsurface pool of pco2 look like prior to these events? Is that getting efficiently drawn down or is the biological response week and so the physical transport is the main control over the surface carbon concentration? See more discussion on the role of event based air-sea carbon fluxes in annual variability for a region here: https://agupubs.onlinelibrary.wiley.com/doi/epdf/10.1029/2010JC006625
**Response:**

We have reworded lines 280ff to de-emphasize the upwelling events. As stated above, we do not see this as the main focus of our paper as these are short-term events that have no marked influence on annual or shelf-wide fluxes.

In addition to comparing the annually integrated air-sea flux in the upwelling bin to the shelfbreak bin (see previous response), these lines have now been reworded to (see lines 340ff in revised manuscript; changes are in bold italics): "***Our regional model shows*** that upwelling events ***could be a large contributor to setting*** the $CO_2$ signal in the summer on the inner portion of the Scotian Shelf, act***ing*** to lower $pCO_2$ ***here*** and ***slightly reducing*** outgassing compared to the ***outer*** shelf."

Regarding the comment on phytoplankton growth, we have added the transects below to the supplement (Section S6 and Figure S11) so readers are informed of other variables at the time of the upwelling event. Our interpretation is that during the event, physical transport is the main control on the spatial variability.

In addition, as per the suggestion from the other Reviewer, we have added a Taylor decomposition to better illustrate how these different factors are affecting the $pCO_2$ signal during the upwelling events (see lines 149ff in the Methods, lines 263ff and Figure 7 in the Results, and lines 322ff in the Discussion).

[Figure]

Minor Comments:
Line 52-52: Please add the Feely et al. 2008 citation here (https://science.sciencemag.org/content/320/5882/1490).
**Response:** Yes, we have added this reference (see line 52 in revised text).

The model gas transfer function chosen is Ho et al. (2006), which is different than the earlier Fennel model iterations. How does this choice (between all of the existing gas transfer functions available) influence your results?

**Response:** Since we are not focusing on short-term, high-wind events, most of the gas transfer functions yield similar results without much divergence. A few years ago, we updated from Wanninkhof (1993), which is the gas transfer function originally used in the Fennel model, to Ho et al. (2006) because we were criticized for using on outdated parameterization. Although it is thought that Wanninkhof (1993) potentially overestimates gas transfer, particularly at higher wind speeds (Ho et al. 2006), both gas transfer functions yield similar results for the air-sea $CO_2$ fluxes in our model.

Lines 213-214: Can you add statistics to support "good agreement" here?

**Response:** We have moved the figure that this sentence references to the Supplement and removed this sentence from the main text. However, in Figure 3, we do report overall statistics comparing the model to the Atlantic Condor transect, with an RMSE of 28.7 µatm and a bias of 13.9 µatm.

Line 292: If you averaged your two regions together - would your results be more in line with theirs?

**Response:** No, if we averaged our two regions together our estimate would not be more in line with the estimate from Laruelle et al. (2015).

Line 314: " thermodynamic signal in pCO2 outweighs the influence of biological activity " This is not clearly shown in this work.

**Response:** Agree, this statement is a reference to Shadwick & Thomas (2014). We have modified the text to (see lines 376ff in revised text): *"In summer, temperature-normalized $pCO_2$ continues to decrease rather than follow the increasing temperature signal of non-normalized $pCO_2$. Previous studies have noted that, in summer, the thermodynamic signal in $pCO_2$ outweighs the influence of biological activity (Shadwick et al. 2011; Shadwick & Thomas 2014), which could explain the differences in seasonality between $pCO_2$ and temperature-normalized $pCO_2$ in the present study. We believe this thermodynamic influence is an important factor driving the net outgassing observed on the Scotian Shelf, particularly when combined with the delivery of DIC-rich water from the Labrador Sea."*

Figure 2 - Add statistics (RMSE etc) directly to these plots. Is the smoothing of the model part of the issue? what about the time/spatial mismatch? Is the socat data being interpolated to the location of the mooring? was the model? how was that extracted? These details need to be added to the methods as well – evaluation methods.

**Response:** We have added the RMSE and bias directly to the plots. The model was not smoothed and model and data are shown in the same location (no spatial mismatch). We now focus on only one year from the extended model simulation, but include the shaded area as the range from multiple years to illustrate the temporal variability. We have redone all figures and are now correcting for the long-term trend by mapping values from different years onto the same reference year. We believe the main issue is that the magnitude of the bloom is not large enough in the model to capture the rapid and large decline in $pCO_2$, as stated in the text. The model

output was extracted at the buoy location. The SOCAT data was averaged over the Scotian Shelf, as indicated in the figure caption.

Figure 3 – The summer gradient generated by the upwelling (observed) does not appear to be captured by model. Can you address this with respect to the localized mechanism that is the focus of this work? Please add some discussion of this to the text. Is the time period the same between simulated and observed?
**Response:** Is the Reviewer perhaps referring to Figure 4? Figure 3 is not intended to show evidence of summer upwelling in either observations or the model but shows the annual and shelf-scale changes in $pCO_2$. Upwelling in the model is also illustrated in Figures 5 and 6.

Figure 4 – the longitudinal gradient in the observations does not appear to be well captured by the model. Is there additional evidence that the model simulates the upwelling in this area well?
**Response:** The occurrence of summer upwelling is well-documented on the Scotian Shelf (see some examples from satellite and models below). The intensity of an upwelling event and the width of the coastal band of cold water varies from event to event and is directly related to the strength and duration of the upwelling-favourable wind. Hence upwelling bands are wider in some events than in others. We are not directly comparing the same event in the model as in the observations since the model simulation does not extend to 2018/2019 (the time period of the Condor transect observations), therefore we do not expect the extent of the upwelling area to be the same between the model and observations.

Examples of upwelling events and the associated band of upwelled water on the Scotian Shelf: Petrie et al. (1987) used satellite images of the region to show the development of a band of cool water along the southern shore of Nova Scotia over the month of July 1984 caused by upwelling-favourable winds (see Figure below).

[Figure]

**Figure 1: Satellite infrared imagery of sea surface temperatures from (a) July 7, (b) July 14, (c) July 21, (d) July 25, (e) July 31 and (f) August 6, 1984. Image is from Petrie et al. (1987) illustrating narrow band of cool water on the southern shore of Nova Scotia during a period of upwelling-favourable winds.**

A more recent example from Shan (2016) showing both satellite images and simulated model snapshots of SST in July 2012 is given below and illustrates again the band of cool upwelled waters on the southern shore of Nova Scotia in the vicinity of the coast. Shan (2016) noted two distinct upwelling events during 2012, one that peaked July 22 and the other September 1, 2012.

[Figure]

**Figure 2: MODIS satellite remote sensing data of SST and Chlorophyll concentrations over the central Scotian Shelf and adjacent waters from July 22 and September 1, 2012 (from Shan 2016). Note that the shelf break is outside the frames. 100 m and 200 m isobaths are shown in black and gray contour lines, respectively.**

[Figure]

**Figure 3:** Snapshots of simulated SST over the central Scotian Shelf in July 2012 with instantaneous wind stress vectors plotting as black arrows (DalCoast-CSS model from Shan 2016).

These references further illustrate that some upwelling events create larger bands of upwelled water along the coastline, such as in Figure 1 and panel (c) in Figure 2.

Figure 6 - Highlight the "nearshore" region you mention in the text on this figure. The DIC gradient is not as severe as in the CCS. Consider putting it in this space: https://www.sciencedirect.com/science/article/pii/S0278434317303643#f0005
**Response:** Indeed, this upwelling is very different from the upwelling in the CCS (see above comments).

Figure 7 – Please add other parameter time series to this plot including temperature, salinity and most important winds (both modeled and observed).
**Response:** We have added more parameters to this time series, now Figure 5 in the revised manuscript.

Figure 8- More detail needs to be added to methods about how these comparisons were made.
**Response:** We added more detail to the figure caption, which now reads (additions in bold italics) "*Monthly and annual air-sea $CO_2$ flux calculated from the model on the entire Scotian Shelf (pink), **extracted** at the CARIOCA buoy location (black), and from the buoy observations (blue). **Flux is averaged over simulation years 2006-2014 for the model, and years 2007-2014 for the CARIOCA observations. Error bars are +/- 1 standard deviations between years**.*"

Figure 9 – Please add vandemark discussion to the text. What is the far right "section"?
**Response:** We have added Vandemark to the discussion (see line 354). The rightmost section is a "merged" location as both Laruelle papers define a larger area and not solely the Scotian Shelf or Gulf of Maine. We have relabelled this accordingly.

Finally, the title would be more informative if it were about the science question the paper is trying to address.
**Response:** We believe the Reviewer may have misunderstood our intended science question and hope this is clarified by the above responses.

McGarry, K., Siedlecki, S. A., Salisbury, J., & Alin, S. R. (2021). Multiple linear regression models for reconstructing and exploring processes controlling the carbonate system of the northeast US from basic hydrographic data. Journal of Geophysical Research: Oceans, 126, e2020JC016480. https://doi.org/10.1029/2020JC016480

Xu, Y.âY., Cai, W.âJ., Wanninkhof, R., Salisbury, J., Reimer, J., & Chen, B. (2020). Long-Term Changes of Carbonate Chemistry Variables Along the North American East Coast. Journal of Geophysical Research: Oceans, 125, e2019JC015982. https://doi.org/10.1029/2019JC015982

**A modeling study of temporal and spatial pCO2 variability on the biologically active and temperature-dominated Scotian Shelf**

**Response to Comments by Reviewer 2**
(Reviews are included in black font; Responses are in blue font)

The authors use output from a regional oceanic biogeochemical model and mooring/ship-board observations to study the seasonal cycle of surface pCO2 and sea-air CO2 fluxes. The general findings are that the Scotian Shelf acts as a net annual source of CO2 to the atmosphere and that biological activity and temperature are the main drivers of the pCO2 variability. The authors also try to show that coastal upwelling is responsible for low near-shore surface pCO2 in summer. Overall, I find the manuscript well organized. However, I have several concerns (see below) that need to be addressed.
**Response:** We appreciate the constructive comments and have addressed them as described below.

Line 14: Might be good to mention here by how much pCO2 changes due to this steep increase in temperature.
**Response:** Agree, we have modified the text as follows (addition in bold italics; lines 12-15 in the revised manuscript): "*Surface pCO$_2$ undergoes a strong seasonal cycle **with an amplitude of ~200-250 μatm**. **These changes are** associated with both a strong biological drawdown of Dissolved Inorganic Carbon (DIC) in spring **(corresponding to a decrease in pCO$_2$ of 100-200 μatm)**, and pronounced effects of temperature, which ranges from 0ºC in the winter to near 20ºC in the summer**, resulting in an increase in pCO$_2$ of ~ 200-250 μatm.**"

Lines 33-36: Since you specify the type of measurements that suggest that the Scotian Shelf is a net CO2 source, it would be interesting to know what type of data suggests that it is a net sink.
**Response:** It is the same type of measurement (surface ocean pCO$_2$), but different data sets are used in these studies. We have modified the text as follows (addition in bold italics; lines 36-38 in the revised manuscript): "These findings are in contrast to other studies ***using observations from the SOCAT database,*** indicating that the Scotian Shelf follows the global trend and acts as a net sink of CO$_2$ (Laruelle et al. 2014; Laruelle et al. 2015; Signorini et al. 2013)."

Line 63: I would be careful calling any model "accurate"! If the model has been evaluated properly (if the region has an adequate amount of observations), then I bet these studies identified some deficiencies. I would suggest to briefly summarizing the previous model evaluation here and state unknowns due to lack of data, if applicable.
**Response:** Agree, we have modified this as follows (added/modified text in bold italics; lines 64-65 in the revised manuscript):
 "*In the present study, we employ a high-resolution biogeochemical model of the northwest North Atlantic to examine the magnitude, variability and sign of the air-sea CO$_2$ flux on the Scotian Shelf. Previous studies **evaluated** our **model's ability to** represent the physical (Brennan et al. 2016, Rutherford & Fennel 2018) and biological (Laurent et al. 2020) dynamics of the region.*"
We have also added more detail about these evaluations in the methods, where we describe the physical and biological model set up.

Line 105: What are the possible implications of using a river climatology to force the model? Is anything known about interannual or longterm changes to the riverine input?
**Response:** The intended purpose of this paper is to focus on the seasonal variability not interannual or long-term changes. River inputs make up a very small fraction of the water on the Scotian Shelf (see Rutherford & Fennel 2018) and variations in riverine chemistry over this short period would be negligible. We have emphasized throughout the revised manuscript that the focus of this study is on a seasonal scale (see for example lines 11, 67, 68, 310, 371, 396 in the revised manuscript).

Line 102: Again, a brief summary of deficiencies and skills of the model would be good.
**Response:** Agree, we have added the following (addition/modification in bold italics):

Lines 114-119 in the revised manuscript: "*For a detailed description and validation of the biological model, we refer to Laurent et al. (2021), **who compared the model output with glider transects of temperature, salinity and chlorophyll, and in situ measurements of chlorophyll and nitrate. The model was evaluated on a seasonal scale for the entire model domain, mainly in the surface (top 100 m). Laurent et al. (2021) showed that the model outperforms global models for the region for all variables and that the timing of the spring bloom is well represented, but the model slightly underestimates the magnitude of the bloom and tends to slightly overestimate nitrate throughout the year.* "

Lines 107-110 in the revised manuscript: *"Full details on the physical model setup and its validation can be found in Brennan et al. (2016) and Rutherford & Fennel (2018). **These studies have shown that our model simulates the vertical structure and seasonal cycling of temperature and salinity on the shelf well. The model captures mesoscale features and coastal upwelling events, and simulates the volume transport throughout the region in agreement with observation-based estimates.**"*

Line123: Why is it drifting and how does the nudging impact the actual model skill. I was surprised that so much nudging was done for a relatively small model domain. Are the nudged areas not used in the analysis? And if these areas are used, how do you deal with them? Would be helpful to show the nudged areas in Figure 1.
**Response:** The nudging zones along the boundary are not used in the analysis, which we have stated explicitly in the revised manuscript. It is common to apply boundary nudging in regional domains as a method to impose low-frequency variability from outside the domain. The nudging timescale is long (60 days at the boundary, linearly decreasing to zero at the inner edge of the nudging zone). This means that nudging is weak. Since the internal dynamics of the Labrador Sea, which set the seasonal cycle of physical and biogeochemical conditions at the northeastern boundary, are not represented in our regional shelf-focussed domain, boundary nudging is applied essentially to impose information from outside the domain in a band along the model's open boundary. This benefits model skill by eliminating unrealistic drifts.
We have updated the manuscript as follows: (Lines 130-134 in the revised manuscript, added text is in bold italics): "*DIC is nudged in an 80-grid-cell wide buffer zone along the eastern boundary, with nudging linearly decaying away from **a nudging timescale of 60 days at** the boundary to a value of 0 in the 81st grid cell. At all other boundaries, a 10-grid buffer zone is*

*used, as with temperature and salinity.* **Use of a wider boundary nudging zone along the eastern boundary was found to be beneficial in imposing low-frequency variability from the Labrador Sea at the northeastern boundary. The nudging zones are not used in the analysis.**

Line 131: Model spin-up of a biogeochemical model usually takes 6-10 years. Can you show that 1 year is enough and the model won't drift anymore? For example run the model for 10 years perpetually, using the 2000 conditions. Does DIC remain relatively stable, without drifting?
**Response:** We have now run a longer simulation (1999-2014), which has been analyzed and replaced the previous 6-year simulation (1999-2004) from the original manuscript. We now focus on the years 2006-2014 of the longer simulation, which encompass the observation years. In addition, we have included figures in the supplement (see supplement section S2 and Figures S2-S5) with a timeseries of surface $pCO_2$ on each of our shelves for the years 2000-2014 to illustrate the interannual variability in the model and show that there is no noticeable model drift in these years.

Below is a figure of surface $pCO_2$ averaged on the Scotian Shelf and locally at the CARIOCA buoy comparing year 1999 to year 2000 (i.e. years 1 and 2 of the simulation) to illustrate the model spin-up. Model spin-up is seen mainly within the first 75 days (or first 3 months) where $pCO_2$ is lower and relatively constant in 1999 compared with 2000 (and all other subsequent years, see Figure S2 and S3). This spin-up period aligns with the residence time on the Scotian Shelf of ~ 3 months (see Rutherford and Fennel 2018).

[Figure]

Line 147: Need to label the location of the Halifax and Deep Panuke gas platform.
**Response:** Agree, we have added this to Figure 1 and updated the text as follows (additions in bold italics, see lines 176-177 in revised manuscript): "*The ship transits weekly to biweekly between the Halifax* **Harbour (Bin 1)** *and the Deep Panuke gas platform off Sable Island* **(Bin 2)** *on the Scotian Shelf (Figure 1)."*

Line 163: "from top to bottom…" belongs into caption and not into main text. Also, describe method you used to temperature normalize pCO2 in caption.
**Response:** Agree, we have modified this accordingly.

Line 164: To me it is confusing to talk about days and months. I would just stick to months, since days are less obvious – The reader would have to first convert to the month before understanding what time of the year you are referring to. I don't see how pCo2 is relatively constant between day 0

to 75. Are you referring to the temp normalized pCO2? But even temp normalized pCO2 is increasing during this time. Might be better to give a range here?

**Response:** We prefer using the day of year for this section because it is more specific than referring to months and believe it is clear.

Line178: add "buoy" to "… at the low end of the **buoy** observations

**Response:** Agree, this change was made.

Line 182: I don't think the word "consistent" is appropriate here? The model seems to underestimate the DIC drawdown due to primary production compared to both types of observations (temp norm. pCO2).

**Response:** Agree. We have modified this line to (now 216-217; modification in bold italics): *"The bloom-related minimum in pCO₂ in the model is approximately 50-75 µatm higher than the buoy observations **and approximately 25-50 µatm higher than** the Atlantic Condor observations."*

Line 189: verb is missing.

**Response:** This has been corrected, and now reads as follows (see 224-225 in revised manuscript, changes in bold italics): *"The model tends **towards** slightly higher pCO₂ across the shelf **compared to** the ship data, **but the bias along the ship track is about half the magnitude as that at the buoy.**"*

Line 209: Figure 4 shows how the model struggles to simulate the spatial variability, which should be pointed out.

**Response:** We have added a sentence at the end of the first paragraph stating that the model does not show the small-scale variations in pCO₂ that are seen in the Condor transect. However, we would like to add that this is not surprising. It is common that models produce much less variations than underway pCO₂ observations. We would also point out that underway measurements are prone to many errors and that the variations may at least partly be due to measurement artefacts. See lines 240ff in the revised manuscript for this addition: "***Small-scale spatial variability in the observations is not captured by the model, but may, at least in part, be due to measurement artefacts of the underway system.***"

Line 212: add east or west to longitude description

**Response:** Yes, we have added this throughout the paper.

Line 216: I don't think these events are all that obvious in the observations. There were only a total 3 inner shelf observations during this time period, two of which are actually higher than an outer shelf observation point (also the only one during this period). I agree, that this is obvious in the model, but would be more careful with this statement for the observations. I just don't think that the observations can be interpreted that way… Im also not conviced by the proposed mechanism that leads to low pCO2, despite high DIC. What does the salinity profile look like? I think this section needs something like a Taylor decomposition to show that what is responsible for the low pCO2 (see details in

Rheuban, J. E., Doney, S. C., McCorkle, D. C., and Jakuba, R. W.: Quantifying the effects of nutrient enrichment and freshwater mixing on coastal ocean acidification, J. Geophys. Res.-Oceans, 124, 9085–9100, https://doi.org/10.1029/2019JC015556, 2019.

Or

Hauri, C., Schultz, C., Hedstrom, K., Danielson, S., Irving, B., Doney, S.C., Dussin, R., Curchitser, E.N., Hill, D.F, and Stock, C.A.: A regional hindcast model simulating ecosystem dynamics, inorganic carbon chemistry, and ocean acidification in the Gulf of Alaska, Biogeosciences, 17, 3837–3857, https://doi.org/10.5194/bg-17-3837-2020, 2020.
**Response:** The observations with higher $pCO_2$ nearshore have been removed from the analysis, due to measurement system error. There is 1 instance in the observations where this low $pCO_2$ signal is very clear, which is associated with cold temperatures (refer to Figure 5 in the original manuscript; now Figure S9 in the supplement). We agree that with only 1 instance in the observations of low $pCO_2$ nearshore, it is difficult to draw unequivocal conclusions about the mechanisms driving this. We have therefore rephrased this section as using the model to hypothesize what could be driving this low $pCO_2$ nearshore. See the following changes in the revised manuscript:

Add at lines 245ff: "***With more obvious examples in the model than in the observations, we use the model to investigate into a possible explanation for this decreased $pCO_2$ nearshore.***"

And lines 312ff (additions in bold italics): ***"Notable occurrences of spatial variability of $pCO_2$ on the Scotian Shelf occur throughout the summer months in both the model and observations. With only 1-2 clear examples of lower $pCO_2$ within ~ 25 km of shore in the observations, we used our model to hypothesize about a possible mechanism driving this variability. In the model, we found that*** coastal upwelling events are driving ***the summertime*** spatial variability in pCO2 on the Scotian Shelf ***and could explain the variability in the observations as well."***

We additionally have added more variables along the transect to the supplementary info (see section S6 and figure S11). We also have added a Taylor Decomposition (in the revised manuscript, see lines 149ff in the Methods, lines 263ff and Figure 7 in the Results, and lines 322ff in the Discussion).

Line 264: "Accurate" means: "correct in all details; exact"- as menti0oned earlier, I yet have to see a model that can be described as "accurate". I would tone it down… especially because you start the sentence with "This limitation aside…"
**Response:** Agree, we have changed the word "accurately" to "well captured" (see line 310 in revised manuscript).

Line 270: would be nice to calculate how much the temperature change affects pCO2 and how much DIC increases affect pCO2….
**Response:** Agree, we have used the Taylor Decomposition to accomplish this. At line 323ff in the modified text, we have added: "***In the example explored in the present study, the upwelled water comes from ~ 20-25 m depth that has a $pCO_2$ approximately 100 uatm lower than the rest of the shelf. Temperature in the upwelled water is acting to lower $pCO_2$ by ~ 150 uatm whereas DIC is acting to increase $pCO_2$ by ~ 50 uatm compared to the rest of the shelf. If deeper water was being upwelled to the surface, DIC would likely start to be the dominant factor in setting $pCO_2$ during these events (Figure 7)."***

**Figures – I really like the color choices of the figures!**
**Response:** Thank you!

Figure 1: It would be nice to give the reader a better understanding of where the Scotian Shelf is located. Maybe a zoomed-out map as an insert? Label all location names you are mentioning in the paper e.g. Halifax Harbor. What are bin 1 and bin 2? Please describe in caption. Also, LAt and Lon labels are missing, including whether it is north or south, and east or west. This should be adjusted for all figures throughout.
**Response:** We have made these changes as suggested.

Figure 2: Correct "Glider obser**v**ations"
Identify grey band in legend for consistency.
What are the two different x-axis?
**Response:** We have made these changes. We included a DOY and month x-axis to help the reader (since we refer to DOY in the text).

Figure 4: What are these inserts? Zoom in? Does not seem to show what you see in the smaller box below. This figure is kind of confusing. What are we actually looking at? Are there 365/5 lines total per figure?
**Response:** We have made the caption for this figure clearer and more descriptive.

Figure 5: On the left, there is no top and lower panel... please adjust accordingly. Also, maybe identify "thick black line" as "vertical black line"
**Response:** Agree, these have been modified. This figure has also been moved to the supplement since we have expanded Figure 7 (now Figure 5) to replace it.

Figure 6:Please identify the variable that goes with each unit next to the colorbar. Always good to specify units of all variables in the caption too. Also, define abbreviations in all figures e.g. dissolved inorganic carbon (DIC). Figures and captions should be readable without reading the manuscript. Since you refer from figure 5 to this figure, you should mention here that July 11, 2000 is indicated in figure 5. Please show the transect line again in the map and label it with lat and lon. Why not also show a profile of pCO2 here to make the point that pCO2 decreases during upwelling event.
**Response:** Agree. We have made these changes.

Figure 7: Add "the" to …the values from **the** nearshore bin…
**Response:** Done.

Figure 8: What do the error bars mean? What are they based on? What are the numbers behind +/-? 1 STD? Needs to be defined in caption.
**Response:** Error bars are standard deviation between years. We have added this to the figure caption.

Figure 9: Why are some bars faded? Are all ingassing bars faded? Needs to be defined.
**Response:** Yes, all ingassing bars are faded. We have added this to the figure caption.

[revised manuscript text omitted]

---

## Author Response (AR2)

**A modeling study of temporal and spatial $p$CO$_2$ variability on the biologically active and temperature-dominated Scotian Shelf**

(Editor/reviewer comments are included in black font; Responses are in blue font)

**Dear Dr. Rutherford:**

**I am now in receipt of two reviews of your revised version. One referee (#1) has evaluated your initial (BGD) version; the other one #3 is new to this paper. Both agree that your paper has improved very much and recommended eventual publication. However, they identified some (minor) issues to be addressed. Details are given in the report below. Referee # 3 suggested to include some discussion on the role of buffering state. This echoes the need to include the Revelle sensitivity factor expressed by referee #1 during the initial submission. I agree with both referees that this may strengthen your paper, but I leave it up to you. (Your name is on the paper).**

**I am looking forward receiving another revision, that after inspection by me, will likely be accepted for publication. Thank you for considering Biogeosciences,**

**Best regards, Jack Middelburg, Associate Editor**

**Response:** Thank you for your time involved with our paper. As we mentioned in our response to Referee #1 during the initial submission, discussion of buffering state/Revelle factor are outside the intended scope of our paper. Discussion of buffering capacity is of course important; however, we think adding it to our current paper would take away from our main messages and discussion. We have upcoming work which uses the same model and focuses on long-term carbonate chemistry trends, and we feel that might be a more appropriate body of work to discuss regional buffering capacity.

We have outlined below the changes we have made to address the comments from Referee #1. There were no comments/suggestions from Referee #3 in their report, but we thank them for taking the time to review our paper.

**Suggestions from Referee #1:**

**The authors did a great job with clarifying and updating the methods in this version of the manuscript. The clarification about the science questions in the text is a welcome addition as well. While the authors spend a lot of time on the discussion of Figure 9 - and rightly so - the reader is still left with questions. One important one that could be cleared up quickly in the text or in the figure is about the last panel (SS +GOM) - couldn't this also be compared to (g) the model output? Is there a reason the models would agree more with the latest round of the SOCAT products? Are the "newly" included data sets in a region that is important to**

**getting the direction of the regional flux correct? The addition of this detail would help clarify this disagreement and help modelers in the area understand what is needed to properly simulate this flux.**

**Response:** Thank you for taking the time to review our paper again. We have addressed your remaining comments about Figure 9 as detailed below.

Firstly, we have updated Figure 9 to include a model estimate of GoM + SS to compare more directly with Laruelle et al. 2014 and Laruelle et al. 2015 (the last panel of Figure 9; see below).

[Figure]

Secondly, regarding the agreement of our model estimates with previous studies, we have added text (bold italics below) to hopefully answer the reader's remaining requestions.

**Lines 350ff:** According to the model, the Scotian Shelf acts as a net source of $CO_2$ to the atmosphere (+1.7 ± 0.2 mol C m$^{-2}$ yr$^{-1}$), the Gulf of Maine is a net sink of $CO_2$ (-0.5 ± 0.2 mol C m$^{-2}$ yr$^{-1}$) and the Grand Banks act as a net sink of $CO_2$ (-1.3 ± 0.3 mol C m$^{-2}$ yr$^{-1}$). These results are in agreement with Shadwick et al. (2014) for the Scotian Shelf, and Laruelle et al. (2014, 2015) for the Gulf of Maine. Our results disagree, however, with results from other global (Laruelle et al. 2014) and regional studies (Laruelle et al. 2015; Signorini et al. 2013; Vandemark et al. 2011). The discrepancy in reported air-sea $CO_2$ flux between these studies is partly a result of how each study defines the area of the Scotian Shelf and Gulf of Maine. For example, Laruelle et al. (2015) calculates one estimate for both the Scotian Shelf and Gulf of Maine. The shelves of eastern North America are diverse, particularly in width and circulation features, and defining

them as a single region is not representative. Additionally, the Scotian Shelf waters are strongly influenced by cold, carbon-rich Labrador Sea water, which is not the dominant endmember south of the Gulf of Maine (Loder et al. 1998, Rutherford & Fennel 2018; Fennel et al. 2019). Calculating a single flux estimate for the entirety of this dynamically diverse region is problematic and will yield a different estimate than when considering smaller and more specific regions. However, this only partially explains the difference in flux estimates.

Another reason is that the global SOCAT database was missing important regional data until recently. Signorini et al. (2013) used data from version 1.5 and Laruelle et al. (2014, 2015) used data from version 2.0 of the SOCAT database. Neither of the observational datasets used in the present study were included in SOCAT versions 1.5 and 2.0. **Error! Reference source not found.** illustrates the difference between different SOCAT versions for seasonal $p$CO$_2$ on the Scotian Shelf. SOCAT v2020 has consistently higher average $p$CO$_2$ values than v1.5 and v2, with at least double the number of years and a much larger number of observations going into each monthly average (on the order of 1000 to 10000 measurements in v2020 versus 100 to 1000 in v1.5 and v2). We believe that flux estimates using the updated SOCAT v2020 will agree better with our estimates and those of Shadwick et al. (2014) *since SOCAT v2020 includes more observations with higher spatial and temporal resolution to better capture the distinct seasonal cycle here. Our study, however, only focuses on the recent seasonality of pCO₂, making it difficult to distinguish if earlier SOCAT versions miss the regional dynamics solely due to low resolution of observations, or if the estimates from the different SOCAT versions are reflective of a shift in the behaviour of the shelf system. More work should therefore be done to better understand how variability on longer timescales could be affecting regional pCO₂ and if that variability could also be a reason for the disagreement between the different SOCAT version.*

**No comments from Referee #3.**